# Towards Interaction Detection Using Topological Analysis on Neural Networks

**Zirui Liu**
Dept. of Computer Science
Texas A&M University
College Station, TX
tradigrada@tamu.edu

**Qingquan Song**
Dept. of Computer Science
Texas A&M University
College Station, TX
song_3134@tamu.edu

**Kaixiong Zhou**
Dept. of Computer Science
Texas A&M University
College Station, TX
zkxiong@tamu.edu

**Ting-Hsiang Wang**
Dept. of Computer Science
Texas A&M University
College Station, TX
thwang1231@tamu.edu

**Ying Shan**
Tencent
Beijing, China
yingsshan@tencent.com

**Xia Hu**
Dept. of Computer Science
Texas A&M University
College Station, TX
hu@cse.tamu.edu

## Abstract

Detecting statistical interactions between input features is a crucial and challenging task. Recent advances demonstrate that it is possible to extract learned interactions from trained neural networks. It has also been observed that, in neural networks, any interacting features must follow a strongly weighted connection to common hidden units. Motivated by the observation, in this paper, we propose to investigate the interaction detection problem from a novel topological perspective by analyzing the connectivity in neural networks. Specially, we propose a new measure for quantifying interaction strength, based upon the well-received theory of persistent homology. Based on this measure, a **P**ersistence **I**nteraction **D**etection (PID) algorithm is developed to efficiently detect interactions. Our proposed algorithm is evaluated across a number of interaction detection tasks on several synthetic and real world datasets with different hyperparameters. Experimental results validate that the PID algorithm outperforms the state-of-the-art baselines.

## 1 Introduction

Statistical interaction describes a subset of input features that interact with each other to have an effect on outcomes. For example, using Phenelzine together with Fluoxetine may lead to serotonin syndrome [1]. Interaction detection problem is to quantify the influence of any subset of input features that may potentially be an interaction. The quantified influence in the problem is called interaction strength. With detected interactions, we may formulate hypotheses that could lead to new data collection and experiments. Traditional methods often need to conduct individual tests for all interaction candidates [2, 3] or pre-specify all functional forms of interests [4, 5]. Recent efforts have been dedicated to extracting learned interactions in neural networks by designing measures for quantifying interaction strength based on predefined conditions in a heuristic way [6]. It has been shown to be an effective way to detect interactions and avoid the drawbacks of traditional methods.

One key observation in the state-of-the-art methods is that any interacting features must follow strongly weighted connections to a common hidden unit before reaching the final output layer [6, 7]. Based on this, the strength of interactions can be modeled by the connectivity between these interacting features and output units of a trained neural network. This motivates us to solve the problem from a novel topological perspective. Specifically, our framework builds upon computational

techniques from algebraic topology, specially the persistent homology, which has been shown beneficial for several deep learning models [8–10]. The main advantages of utilizing persistent homology are twofold. First, it provides us a rigorous mathematical framework for analyzing the connectivity in a trained neural network. Second, persistent homology can be used to quantify the importance of each connected component in the neural network, and the connectivity between interacting features and output units are characterized by these connected components.

However, persistent homology cannot be directly applied to quantify the interaction strength from the importance of connected components that link interacting features to units in the final output layers. Also, an interaction is a subset of input features, which is not within the scope of persistent homolgy. The key challenge remains to define a measure for quantifying the interaction strength, which should provide meaningful insights while maintaining theoretical generality.

In this paper, we show that the key concepts of *persistence diagrams* in persistent homology theory can be extended to interactions for tackling the challenges. Specifically, we propose a new measure for quantifying interaction strength, which is computed to reflect the connectivity between interacting features and output units in a neural network. Based on the measure, we propose Persistence Interaction Detection (PID), a framework that can efficiently extract interactions from neural networks. We also prove that our framework is locally stable, meaning that PID is not sensitive to the perturbation of weights in neural networks. Formally, our contributions are as follows:

- We formulate the interaction detection problem as a topology problem. Based on the persistent homology theory, we propose a new measure for quantifying interaction strength by analyzing the topology of neural networks. We then provide analysis for the measure from different perspectives.

- We derive an efficient algorithm to calculate the proposed interaction strength measure. Also, we theoretically analyze the local stability of our proposed framework.

- The proposed PID framework demonstrates strong performance across different tasks, network architectures, hyperparameter settings, and datasets.

## 2 Preliminaries

We first introduce the notations and give the formal definition of feature interactions. Based on the notations, we introduce concepts of the filtration and persistence diagrams. We then show how to build filtration for neural networks in 2.2, serving as the preliminary of our proposed method.

### 2.1 Problem Formulation and Notations

We denote vectors with boldface lowercase letters (e.g., $\mathbf{x}$, $\mathbf{w}$), matrices with boldface capital letters (e.g. $\mathbf{W}$), and scalars with lowercase letters (e.g., $a$). We use $x_i$ to represent the $i$-th entry of vector $\mathbf{x}$, and $W_{ij}$ to denote the entry in the $i^{\text{th}}$ row and $j^{\text{th}}$ column of $\mathbf{W}$. The transpose of a matrix or a vector is denoted as $\mathbf{W}^\top$ or $\mathbf{x}^\top$. For a set $\mathcal{S}$, its cardinality is denoted by $|\mathcal{S}|$. We use $\mathcal{S}\backslash i$ to denote the set $\{j|j \in \mathcal{S} \text{ and } j \neq i\}$. Let $\mathbf{x} \in \mathbb{R}^d$ be the feature vector. An interaction $\mathcal{I}$ is a set of interacting features, where $|\mathcal{I}| \geq 2$. A $K$-order interaction $\mathcal{I}$ satisfies $|\mathcal{I}| = K$. A high-order interaction is an interaction whose order $\geq 3$. We will write $\mathbf{x}^{\mathcal{I}} \in \mathbb{R}^{|\mathcal{I}|}$ as the feature vector selected by $\mathcal{I}$.

Consider a feed-forward neural network (FNN) with $L$ hidden layers (e.g., an MLP). Let $p_l$ be the number of hidden units at the $l^{\text{th}}$ layer. The input features are treated as the $0^{\text{th}}$ layer and $p_0 = d$ is the number of input features. The $l^{\text{th}}$ layer weight matrix is denoted by $\mathbf{W}^{(l)} \in \mathbb{R}^{p_{l-1} \times p_l}$. Given a FNN with weights $\{\mathbf{W}^{(i)}\}_{i=1}^L$, its equivalent weighted directed acyclic graph $\mathcal{G}(V, E)$ can be constructed as follows: We create a vertex for each hidden unit in the neural network and consequently the set of all vertices: $V = \{v_{l,i}|\forall\ l, i\}$, where $v_{l,i}$ represents the $i^{\text{th}}$ hidden unit at the $l^{\text{th}}$ layers; and we assign weight $W_{i,j}^{(l)}$ to each edge in $E = \{(v_{l-1,i}, v_{l,j})|\forall l, i, j\}$.

In this work, we focus on detecting non-additive interactions. The non-additive interaction is formally defined in Definition 1. We remark that detecting "additive interactions" is a trivial task because any "additive interactions" can be decomposed to the sum of two terms, and non-additive interactions are those which cannot be further decomposed.

**Definition 1** (Non-additive interactions [3, 11]). *Let $\{0, ..., d-1\}$ denotes the input feature set. Given a function $f: \mathbb{R}^d \to \mathbb{R}$ and a feature vector $\mathbf{x} = (x_1, ..., x_d)$, $f$ shows no non-additive interaction of $\{x_i, x_j\}$ if $f$ can be expressed as the sum of two functions, $f_{\setminus i}$ and $f_{\setminus j}$, where $f_{\setminus i}$ is a function which does not depend on $x_i$ and $f_{\setminus j}$ is a function which not depend on $x_j$:*

$$f(\mathbf{x}) = f_{\setminus i}(\mathbf{x}^{\{0,...,d-1\}\setminus i}) + f_{\setminus j}(\mathbf{x}^{\{0,...,d-1\}\setminus j}).$$

For example, in the function of $\pi^{x_0 x_1} + \log(x_1 + x_2 + x_4)$, there is a pairwise interaction $\{0, 1\}$ and a 3-order interaction $\{x_1, x_2, x_4\}$. In contrast, $\{x_0, x_1, x_2, x_4\}$ is a spurious interaction. The goal of interaction detection algorithms is to map models into a set of their learned interaction candidates associated with interaction strength. Ideally, a larger value of interaction strength should indicate the true interaction instead of a spurious interaction.

## 2.2 Persistent Homology on Neural Networks

Persistent homology is an algebraic method for identifying the most prominent connectivity characterizing a geometric object, which is widely used in medical imaging and geometric modeling [12, 13]. In this paper, the object we studied is a weighted directed graph $\mathcal{G}(V, E)$ corresponding to a trained feed-forward neural network. In topology, connected components represent the connectivity of the graph. We can apply persistent homology theory to quantify the importance of each connected component in $\mathcal{G}$. To be specific, $(\mathcal{G}, \phi)$ is called a *size pair* [14], where $\phi$ is a *measureing function* [15]. The role of $\phi$ is to take into account the connective properties of $\mathcal{G}$. The $\lambda$-threshold set of $(\mathcal{G}, \phi)$ is defined as follows:

$$L^\lambda = \{x | x \in E, \phi(x) \geq \lambda\},$$

where $x$ is the edge of $\mathcal{G}$. The measuring function $\phi : E \to \mathbb{R}$ maps a specific edge to a real number.

**Definition 2** (Filtration [16]). *Without loss of generality, suppose $\lambda_1 > \lambda_2$, if the corresponding threshold sets satisfy $L^{\lambda_1} \subseteq L^{\lambda_2}$, then $\phi$ is non-decreasing over $\mathcal{G}$. Given $(\mathcal{G}, \phi)$, where $\phi$ is non-decreasing over $\mathcal{G}$, and a set of thresholds follow $\lambda_0 \geq \lambda_1 \geq ... \geq \lambda_n$, the collections of threshold sets $L^{\lambda_0} \subseteq L^{\lambda_1} \subseteq ... \subseteq L^{\lambda_n}$ is called a **filtration** of $(\mathcal{G}, \phi)$.*

We propose to build the filtration for FNNs and define the measuring function as follows. Let $\mathcal{W}$ be the set of weights. Given $\mathcal{W}$ of a trained feed-forward neural network such that $w_{max} := \max_{w \in \mathcal{W}} |w|$ and $\mathcal{W}' := \{|w|/w_{max}|w \in \mathcal{W}\}$, where $\mathcal{W}'$ is indexed in non-ascending order, namely $1 = w_0' \geq w_1' \geq ... \geq w_n' \geq 0$. The weights associated with edges reflect the connectivity between vertices in the networks. Similar to [17], the measuring function $\phi$ for $\mathcal{G}$ is defined as $\phi((v_{l-1,i}, v_{l,j})) = |W_{i,j}^{(l)}|/w_{max}, \forall (v_{l-1,i}, v_{l,j}) \in E$, which represents the edge strength. The sorted weights are used as $\lambda$ in Definition 2. Consequently, the filtration can be constructed as $\mathcal{G}^{w_0'} \subseteq \mathcal{G}^{w_1'} \subseteq ...$, where $\mathcal{G}^{w_i'} = (V, \{(u, v)|(u, v) \in E \wedge \phi((u, v)) \geq w_i'\})$. We remark that $\mathcal{G}^\lambda$ is both a subgraph of $\mathcal{G}$ and a $\lambda$-threshold set of size pair $(\mathcal{G}, \phi)$. $\mathcal{G}^{w_0'}$ is the sub-graph with exact one edge which has greatest weight. As shown in Figure 1, when the thresholds are decreased, edges are added into the sub-graph and vertices will be connected. It is summarized in Figure 1.

The interpretation of $\mathcal{G}^\lambda$ is that, $\mathcal{G}^\lambda$ is the image of $\mathcal{G}$ at different spatial resolution. Edges with larger weights, which indicates stronger connectivity, will appear over a wide range of spatial scales. As the threshold of filters decreases, edges with smaller weights, which indicates weaker connectivity, will start to pass the filter and provide detailed information of $\mathcal{G}$. In the filtration process, these gradually added edges will form different connected components. From persistent homology theory, persistent connected components, which are detected over a wide range of spatial scales, are more representative for the connectivity pattern of $\mathcal{G}$ [18]. Based on this, persistence diagram is a computational tool for quantifying the importance of these emerged connected components.

Given the size pair $(\mathcal{G}, \phi)$, when we decrease $\lambda$, connected components can be *created* (new edges are added, forming new components) or *destroyed* (two connected components joining together). For each connected component $i$, the threshold causes the birth of $i$ is called the *birth* time $b_i$ and the threshold causes the death of $i$ is called the *death* time $d_i$. The persistence diagram tracks these changes and represents creation and destruction of $i$ as a tuple $(b_i, d_i)$. It quantifies the importance of each connected component by its lifetime (persistence).

**Persistence Diagrams**    Given the filtration of a size pair $(\mathcal{G}, \phi)$, with the *birth* time $b_i$ and the *death* time $d_i$ of each connected component $i$ appearing in the filtration, the collection of the *birth* time and

the *death* time tuple $\mathcal{D} = \{(b_i, d_i) | \forall i \text{ appears in the filtration}\}$ is called the **persistence diagrams** of $(\mathcal{G}, \phi)$. The **persistence** of $i$ is $\mathrm{per}(i) = |b_i - d_i|$.

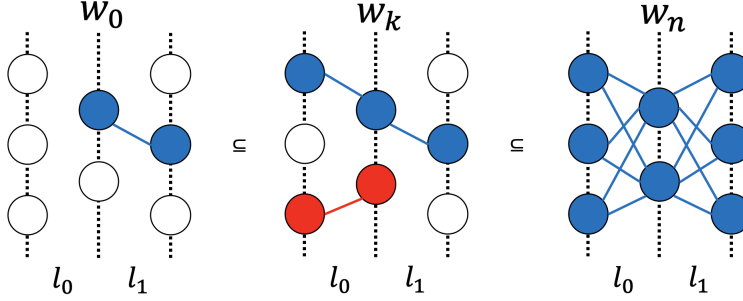

Figure 1: The filtration of a network with two layers. The color scheme illustrates the connected components. The filtration process is represented by colouring connected components that are created or merged when the respective weights are greater than or equal to the threshold $w_i$.

# 3 Persistence Interaction Detection

In this section, we present the proposed PID framework for detecting feature interactions in neural networks. The key intuition of our proposed method is to formulate the interaction detection as a topology problem. That is to find the long-lived connectivity between interactions and outputs in the neural network over a wide range of spatial scales. Based on this, we propose a new measure for quantifying interaction strength (section 3.1). Subsequently, we derive our proposed Persistence Interaction Detection algorithm for calculating the measure efficiently (section 3.2). We then give the stability analysis for our proposed algorithm (section 3.3). To avoid creating confusion in terminology, by "interaction", we mean a subset of input features that satisfies Definition 1 in the rest of the paper.

## 3.1 Persistence As a Measure For Interaction Strength

The concepts of the *birth* time and the *death* time are originally defined for connected components in persistent homology theory. However, the persistence of connected components only implies the importance of themselves. For a particular interaction $\mathcal{I}$, we cannot directly obtain the importance of $\mathcal{I}$ from the persistence of connected components that contain $\mathcal{I}$. Also, an interaction $\mathcal{I}$ is a subset of input features that is not within the scope of persistent homolgy. In this subsection, we will extend these concepts to interactions for deriving a new measure for quantifying interaction strength.

From Definition 1, an interaction is a set of associate features that have an effect on the output. Inspired by persistent homology, we can model this effect by the connectivity between interactions and output units in neural networks. Informally, the *birth* time of an interaction is when there exists a path connecting it to the final output layer, and the *death* time is when the path is also connected to any additional input feature in the filtration. After extending concepts of the $birth$ time and the $death$ time to interactions, we can obtain persistence diagrams of interactions and the interaction strength can be quantified from the lifetime of the connectivity. We first give the definition for the connectivity strength between the interactions and the units in the final output layer in Definition 3. Based on the quantified connectivity, we formally define the *birth* time and the *death* time of an interaction.

**Definition 3** ($\langle\phi = \lambda\rangle$-connected)**.** *Let $(\mathcal{G}, \phi)$ be the corresponding size pair and $\mathcal{G}^{w'_0} \subseteq \mathcal{G}^{w'_1} \subseteq ...$ be the filtration of a neural network, respectively; and $\{0, ..., d-1\}$ denotes the set of input features. For a feature subset $\mathcal{I}$ and a real-number threshold $\lambda$, we call $\mathcal{I}$ and the final output units are $\langle\phi = \lambda\rangle$-connected if: first, there exists a connected component $A \subseteq \mathcal{G}^{\lambda}$ containing $\mathcal{I}$ and the final output units; second, for any such connected component $A$, $\forall i \in \{0, ..., d-1\}\backslash\mathcal{I}$, it satisfies $i \notin A$.*

**Persistence diagrams of interactions** Given the threshold $\lambda_b$, suppose: The feature subset $\mathcal{I}$ and the final output unit are $\langle\phi = \lambda_b\rangle$-connected and, $\forall \lambda_i \geq \lambda_b$, $\mathcal{I}$ and the final output unit are not $\langle\phi = \lambda_i\rangle$-connected, then we call $\lambda_b$ the *birth* time of $\mathcal{I}$. Correspondingly, the *death* time $\lambda_d$ of $\mathcal{I}$ is that $\forall \lambda_i \leq \lambda_d$, $\mathcal{I}$ and the outputs become not $\langle\phi = \lambda_i\rangle$-connected, i.e., interaction $\mathcal{I}$ no longer exists

due to the addition of other input features. The collection of the *birth* time and the *death* time tuple $\mathcal{D} = \{(b_\mathcal{I}, d_\mathcal{I}) | \forall \mathcal{I} \subseteq \{0, ..., d-1\}\}$ is called the **persistence diagrams** of interactions.

After defining the *birth* time and the *death* time of an interaction $\mathcal{I}$, we can quantify its interaction strength by its persistence. We remark that the aforementioned process creates new interaction candidates by associating new features with existing interaction candidates. Some interaction candidates might never born. An example of the persistence of interactions is illustrated in Figure 2.

Let $\mathbf{x} = [x_1, x_2, x_3, x_4]$. $y = x_1^{x_2} + \frac{x_3 x_4}{1000}$. We train a neural network $f$ to minimize the loss $\mathcal{L}(f(x), y) + \mathcal{R}(f)$, where $\mathcal{L}$ is the mean square error and $\mathcal{R}$ is the regularization term. Suppose $w_0' \geq ... \geq w_9'$ are the top ten largest weights in $\mathcal{W}'$. Then the interaction $\{x_1, x_2\}$ and $y$ are $\langle \phi = w_3' \rangle$-connected because of the connected component marked in red. The birth time and the death time of $\{x_1, x_2\}$ are $w_3'$ and $w_6'$, respectively. And the death time of $\{x_1, x_2\}$ marks the birth time of $\{x_1, x_2, x_3\}$.

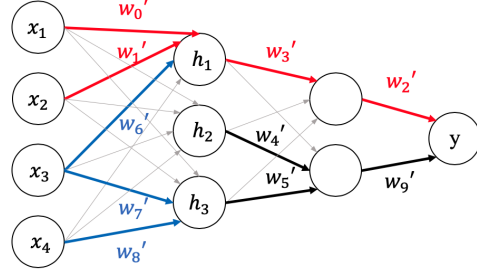

Figure 2: An example for illustrating persistence of interactions.

Intuitively, strength can be quantified in terms of the minimal "amount of change" necessary to eliminate a learned interaction in NNs. The "amount of change" is referred to the distance between changed weights and original weights. From this perspective, the proposed measure is the minimal "amount of change" to eliminate the $\langle \phi = \lambda \rangle$-connectivity between interactions and outputs. For example, if we change $w_6'$ to be as large as $w_1'$, then the input feature $x_2$ and $x_3$ will be simultaneously added to $\{x_1\}$ to form $\{x_1, x_2, x_3\}$, thus $\{x_1, x_2\}$ will never born. The persistence of $\{x_1, x_2\}$ is the gap between the red connected component's smallest weight and $w_6'$ (i.e., $|w_3' - w_6'|$). This gap is the minimal amount of change to eliminate the $\langle \phi = w_3' \rangle$-connectivity between $\{x_1, x_2\}$ and $y$.

## 3.2   Ranking Interactions Using PID

In this subsection we present our PID framework that calculates the proposed measure efficiently. To detect these $\langle \phi = \lambda \rangle$-connectivity between interactions and outputs in Definition 3. We define the mask matrix $\mathbf{M}_{(l)}^\lambda \in \mathbb{R}^{p_{l-1} \times p_l}$ for the $l^{\text{th}}$ layer as

$$[\mathbf{M}_{(l)}^\lambda]_{i,j} = \begin{cases} 1, & \text{if} \quad \phi((v_{l-1,i}, v_{l,j})) \geq \lambda. \\ 0, & \text{otherwise}. \end{cases} \tag{1}$$

The aggregated mask matrix $\mathbf{M}^\lambda \in \mathbb{R}^{p_L \times d}$ are defined as:

$$\mathbf{M}^\lambda = (\mathbf{M}_{(L)}^\lambda)^\top \cdot (\mathbf{M}_{(L-1)}^\lambda)^\top \cdots (\mathbf{M}_{(1)}^\lambda)^\top. \tag{2}$$

**Lemma 1** (Proof in Appendix B). *Let $\{0, ..., d-1\}$ denotes the input feature set, and $\mathbf{M}^\lambda$ denotes the aggregated mask matrix corresponding to threshold $\lambda$, where the $r^{\text{th}}$ row of $\mathbf{M}^\lambda$ is denoted as $\mathbf{m}_r^\lambda \in \mathbb{R}^d$. The feature subset $\mathcal{I}$ and the corresponding $r^{th}$ unit at the final output layer are $\langle \phi = \lambda \rangle$-connected if all elements in $[\mathbf{m}_r^\lambda]^\mathcal{I} \in \mathbb{R}^{|\mathcal{I}|}$ are non-zero and all other elements in $[\mathbf{m}_r^\lambda]^{\{0,...,d-1\} \setminus \mathcal{I}}$ are zero, where $[\mathbf{m}_r^\lambda]^\mathcal{I}$ is the subvector of $\mathbf{m_r^\lambda}$ selected by $\mathcal{I}$.*

As pointed out in [19], different neurons are activated by different patterns (patterns are exactly interactions of raw input features). This indicates that we should generate interaction candidates for each neuron separately. With Lemma 1, we can detect the $\langle \phi = \lambda \rangle$-connectivity between interactions and units in the output layer. However, only care the $\langle \phi = \lambda \rangle$-connectivity between them will ignore the difference between neurons. For example, in Figure 2, all neurons share common interaction candidates in the aforementioned process. The edges gradually added by the filtration process sequentially create the interaction candidates $\{x_1, x_2\}$, $\{x_1, x_2, x_3\}$ and $\{x_1, x_2, x_3, x_4\}$. $\{x_3, x_4\}$ will not be considered because $x_3$ has been merged with $\{x_1, x_2\}$ when they meet at $h_1$. But clearly, $x_3$ and $x_4$ might be a potential interaction candidate because the activation pattern of $h_3$ is largely determined by $x_3$ and $x_4$. To generate interaction candidates for each neuron at a particular layer $l$,

we decompose $\mathbf{M}^{\lambda}$ into $\mathbf{M}_{(l)}^{\lambda_{up}} \in \mathbb{R}^{p_L \times p_l}$ and $\mathbf{M}_{(l)}^{\lambda_{down}} \in \mathbb{R}^{p_l \times d}$, where

$$\begin{cases} \mathbf{M}_{(l)}^{\lambda_{up}} = (\mathbf{M}_{(L)}^{\lambda})^{\top} \cdots (\mathbf{M}_{(l)}^{\lambda})^{\top}. \\ \mathbf{M}_{(l)}^{\lambda_{down}} = (\mathbf{M}_{(l-1)}^{\lambda})^{\top} \cdots (\mathbf{M}_{(1)}^{\lambda})^{\top}. \end{cases} \tag{3}$$

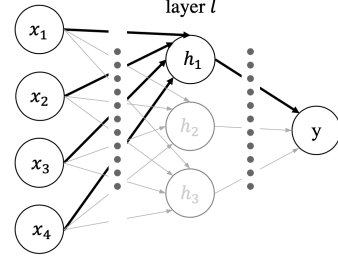

Figure 3: Illustration of PID.

We can obtain the connectivity between a particular neuron $r$ at layer $l$ and units in the final output layer from $\mathbf{M}_{(l)}^{\lambda_{up}}$ (by viewing the layer $l$ as the input layer in the Lemma 1). Similarly, the connectivity between the neuron $r$ and input features can be inferred from $\mathbf{M}_{(l)}^{\lambda_{down}}$. For each neuron $r$ at layer $l$, we generate interaction candidate $\mathcal{I}$ for $r$ only if: first, $\mathcal{I}$ and $r$ are connected; second, $r$ and units in output layer are connected. It is summarized in Figure 3. For example, in Figure 2, if the layer $l$ is set to the first layer, then PID will first generate $\{x_3, x_4\}$ for $h_3$ because $\{x_3, x_4\} - h_3 - y$ are connected once the threshold achieves $w_9'$. For the same interaction candidate generated at different neurons, we aggregate their persistence. We list the full algorithm in Appendix A. From the topological perspective, we model how interactions influence a particular neuron at layer $l$, as well as how this neuron influences units in the final output layer by the quantified connectivity between them. $p$ in algorithm 1 is the norm of the persistence diagram. The $p$-norm is known to be a stable summary for persistence diagrams [20].

### 3.3 Stability of Persistence Interaction Detection

An interaction detection algorithm should not be sensitive to the perturbation of weights, e.g., training neural networks with one or two extra epochs should only change the proposed interaction strength a little. We call that these insensitive algorithms are locally stable. It should be noted that local stability is a necessary condition for fidelity. If the algorithm gives totally different results after training one extra epoch, we cannot tell which one is correct, especially concerning that there are no ground truth labels for interactions in real world datasets. We will show our method is theoretically locally stable.

For two feed forward neural networks $f$ and $g$ with exactly same architecture, let $\mathcal{G}_f(V, E)$ and $\mathcal{G}_g(V, E)$ be their corresponding weighted graph, respectively. We denote the measuring function for $f$ and $g$ as $\phi_f$ and $\phi_g$, respectively. For all interaction candidate $\mathcal{I}$ that are both detected in $f$ and $g$ by Algorithm 1, we denote the interaction strength of $\mathcal{I}$ corresponding to $f$ and $g$ as $\rho_f(\mathcal{I})$ and $\rho_g(\mathcal{I})$ respectively. We propose the following Theorem:

**Theorem 1** (Proof and empirical analysis in Appendix C). *Let $\delta = \max_{e \in E} |\phi_f(e) - \phi_g(e)|$ be the magnitude of perturbation. For all interaction candidate $\mathcal{I}$ that are both detected in $f$ and $g$ by Algorithm 1, it satisfies $|\rho_f(\mathcal{I}) - \rho_g(\mathcal{I})| \leq C\delta$.*

Theorem 1 only states the stability for interaction candidates that are detected by both $f$ and $g$. We note that this is the common situation when the perturbation magnitude is small. However, there might exists the corner case where there are interaction candidates that are only detected in one network, but not the other. We also show the proof that this case only happens if the perturbation magnitude $\delta$ is greater than a threshold in Appendix C.

## 4 Experiments

Our experiments attempt to answer the following research questions: (**Q1**) How effective is PID in detecting true interactions (Section 4.1)? (**Q2**) Is the algorithm sensitive to hyperparameters and different architectures (Appendix E)? (**Q3**) Is considering these detected interactions beneficial for machine learning models (Section 4.2)? (**Q4**) Can PID detect extremely high-order interactions (Section 4.3)? We remark the norm $p$ in Algorithm 1 is set to 2 across all experiments, which captures the Euclidean distance of points in persistence diagrams [20]. The other experiment-specific settings are described in respective sections.

Table 1: AUC of pairwise interaction strengths proposed by PID and baselines on the synthetic functions (Table 4). ANOVA, HierLasso, and RuleFit are deterministic.

|  | ANOVA | HierLasso | RuleFit | AG | NID | PID |
|---|---|---|---|---|---|---|
| $F_1(x)$ | 0.992 | 1.00 | 0.754 | **1±0.0** | 0.985±6.3e−3 | 0.986±4.1e−3 |
| $F_2(x)$ | 0.468 | 0.636 | 0.698 | **0.88±1.4e−2** | 0.776±4.3e−2 | 0.804±5.7e-2 |
| $F_3(x)$ | 0.657 | 0.556 | 0.815 | **1±0.0** | **1.0±0.0** | **1.0±0.0** |
| $F_4(x)$ | 0.563 | 0.634 | 0.689 | **0.999±1.4e-3** | 0.916±6.3e−2 | 0.935±3.9e−2 |
| $F_5(x)$ | 0.544 | 0.625 | 0.797 | 0.67±5.7e-2 | 0.997±8.9e−3 | **1.0±0.0** |
| $F_6(x)$ | 0.780 | 0.730 | 0.811 | 0.64±1.4e-2 | 0.999±3.3e−3 | **1.0±0.0** |
| $F_7(x)$ | 0.726 | 0.571 | 0.666 | 0.81±4.9e-2 | 0.880±2.6e−2 | **0.888±2.8e−2** |
| $F_8(x)$ | 0.929 | 0.958 | 0.946 | 0.937±1.4e-3 | **1.0±0.0** | **1.0±0.0** |
| $F_9(x)$ | 0.783 | 0.681 | 0.584 | 0.808±5.7e-3 | 0.968±2.3e−2 | **0.972±2.9e−2** |
| $F_{10}(x)$ | 0.765 | 0.583 | 0.876 | 1.0±0.0 | **0.989±3.0e−2** | 0.987±3.5e−2 |
| average | 0.721 | 0.698 | 0.764 | 0.87±1.4e-2 | 0.951±7.0e−2 | **0.957±6.2e−2** |

## 4.1 Pairwise Interaction Detection on Synthetic Data

Since there are no ground-truth labels for interactions in real world datasets, to answer **Q1** and **Q2**, we utilize ten synthetic datasets that contain a mixture of pairwise interactions and higher-order interactions, as shown in the Appendix E.1. For higher-order interactions, we tested their pairwise subsets as in [3, 6, 21]. All ten datasets and MLP structures are the same as those in [6]. The detailed experimental settings can be found in Appendix E.1. The pairwise interaction strength of $\{i, j\}$ is obtained by aggregating the strength of all interaction candidates proposed by PID which contain $\{i, j\}$. The layer $l$ in Algorithm 1 is set to the first layer because the neural network naturally separates different interactions in the first hidden layer [6, 7] (see Figure 6 and Figure 7 in Appendix E.2).

We compared the proposed PID with several strong existing algorithms in the interaction detection literature, including ANOVA [2], Hierarchical lasso (HierLasso) [4], RuleFit [11], Additive Groves (AG) [3], and Neural Interaction Detection (NID) [6]. Because both PID and NID detect learned interactions from MLPs in a post-hoc way, we apply the NID and PID on the same MLPs for fair comparison. We ran ten trials of AG, NID, and PID on each dataset and removed two trials with the highest and lowest AUC scores. The AUC scores of interaction strength proposed by baseline methods and PID are shown in Table 1. The heat map of pairwise interaction strength and a detailed analysis about main effects are in Appendix E.2. Here we provide only the general results.

In general, the AUCs of AG and PID are close, except for $F_5$, $F_6$, and $F_8$, where PID significantly outperforms AG. This may be caused by the limitations in the AG's model capacity, which is tree-based [6]. When comparing the AUCs of PID and NID, the AUCs of PID are comparable or better. We note that PID considers connectivity of the entire NN. In contrast, NID leverages weights beyond the first hidden layer to obtain the maximum gradient magnitude of the hidden units in the first hidden layer, loosing some information encoded in latter layers in the process. Hence, the similar results of NID and PID are likely because the latter layers played lesser roles in this specific setting. However, we remark PID constantly outperformed NID with various settings, as shown in Appendix E.3, Figure 8, 9, and 10. To answer **Q2**, we also compare the result based on MLPs with different architectures (Appendix E.3 Figure 8) and regularization strength (Appendix E.3 Figure 10, Figure 9). In general, both NID and PID are insensitive to the architecture of MLPs, and both are sensitive to the regularization strength. A possible reason is that the connectivity between hidden units of a trained MLP is significantly influenced by regularization strength. We show that the AUCs of PID are better than those of NID under all different settings (Appendix E.3).

## 4.2 Automatic Feature Engineering

Intricate feature engineering often plays deterministic roles in winning solutions of Kaggle competitions [22]. In this regard, interaction detection algorithms are invaluable in that they reveal knowledge about the data. A reasonable question is, can different machine learning models benefit from the knowledge to alleviate the need for hand-crafted feature engineering (**Q3**)? We try to answer it by integrating these detected interactions with the original input features and then check the performance gain of models trained on this augmented data.

Table 2: Comparing the quality of features automatically generated by interaction detection algorithms. The "Original" column shows the results of random forest built without using synthetic features.

| Dataset | Original | Random | NID | PID |
|---|---|---|---|---|
| Amazon Employee | 0.8378±0.0046 | 0.7780±0.0575 | 0.8321±0.0299 | **0.8460±0.0079** |
| Higgs Boson | 0.7421±0.0019 | 0.7421±0.0192 | **0.7422±0.0017** | **0.7422±0.0017** |
| Creditcard | 0.9555±0.0390 | 0.9579±0.0377 | 0.9607±0.0333 | **0.9625±0.0354** |
| Spambase | 0.9680±0.0085 | 0.9692±0.0076 | 0.9724±0.0065 | **0.9738±0.0063** |
| Diabetes | 0.8077±0.0334 | 0.8078±0.0335 | 0.8044±0.0335 | **0.8101±0.0349** |

We compare our PID and NID on five real world binary classification datasets. The statistics of these datasets are shown in Appendix F.1 Table 6. Following [23, 24], we explicitly construct synthetic features for each detected interaction candidates and combine these synthetic features with the original feature set. The synthetic feature for interaction $\mathcal{I}$ is the Cartesian product among features in $\mathcal{I}$. The details are described in Appendix F.1. We construct synthetic features for the top ten interactions candidates according to interaction strength. Because of the excellent performance and efficiency of random forest on tabular datasets, similar to [25, 26], we choose the random forest as our learning algorithm. A more detailed experiment setting can be found in Appendix F.1. In Table 2, we also report results of the baseline "Random", where the ten randomly generated synthetic features are used to combine with the original data and then construct the random forest. The AUCs of the random forest construed with different synthetic features are summarized in Table 2.

From Table 2, we remark that incorporating synthetic features generally boost the performance of the random forest model. Namely, NID outperforms Original in the Higgs Boson, Creditcard, and Spambase datasets, while the proposed PID method outperforms all the compared methods. The statistics of detected interactions by different methods are shown in Appendix F.2. Furthermore, we remark that the feature interactions discovered by PID highly coincide with the top solution of the Amazon Employee Challenge [1] (Appendix F.2).

### 4.3 High-order Interaction Detection on Image Datasets

For image data, input features are raw pixels and interactions are patterns that represent visual cues characterizing the object in the image. To answer **Q4**, we apply PID to find out the contributing pattern in a particular image that lead the CNNs to make the prediction. In Section 3, the proposed framework is used for detecting global interactions. Global (or model-level) interaction means the learned interactions for making predictions on the entire dataset. Specifically, the only input to global interaction detection algorithms is the model to be analyzed, without any information about the position or the scale of the object in an image. Local (or instance-level) interaction detection, however, tries to answer what interactions of a data sample lead the model to make a special prediction. We remark that global interaction detection is meaningless for image data because it is not invariant to the position or scales of objects. We show how to extend PID to the CNNs and local interaction detection in Appendix D.

Detecting interactions in a specific image is a more challenging task for the following reasons: First, the order of interactions is extremely high in image data; second, image data is high dimensional by nature. The number of possible interaction candidates grows exponentially with respect to the number of input features, e.g., for a $1 \times 28 \times 28$ image from MNIST dataset, the number of interaction candidates within the search space is $2^{784} \approx 10^{235}$. We note that interactions in an image is similar to the Superpixels [27], which is originally proposed for solving the image segmentation task. However, it is not straight-forward to show detected interaction by considering them as superpixels: First, the interaction in an image is a group of pixels that are not necessarily connected; second, theoretically, each interaction is associated with the interaction

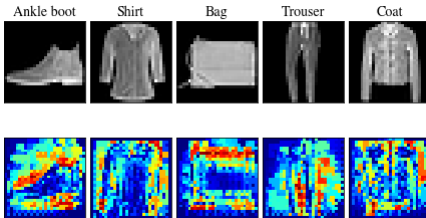

Figure 4: Saliency maps of interaction strength found from applying PID on the CNN trained on FashionMNIST dataset.

[1] https://www.kaggle.com/c/amazon-employee-access-challenge

strength, which cannot be shown by simply breaking the image into different segmentation. To evaluate the detected interactions with better representation of images, instead of connecting pixels that are in the same interaction, we build the saliency map for each input images by visualizing the importance of raw pixels. Specifically, the importance of the raw pixel $i$ is obtained by aggregating the interaction strength of each candidate set which contains $i$.

We trained a simple CNN to classify images on the MNIST dataset [28] and FashionMNIST dataset [29]. The detailed experiment setting can be found in Appendix G. Here we only present the saliency maps of FashionMNIST images. The saliency maps of MNIST images are available in Appendix G. We observe PID is capable of detecting high-order interactions that represent visual cues. From Figure 4, the CNN acquired complex knowledge about the shapes associated with each category. For example, the interpretations of the "Ankle boot" classification show that interaction detection finds the shape of the boot instead of the boot texture. This is indicated by the fact that pixels with higher importance (warm colors) essentially trace the contour of an ankle boot.

## 5    Limitation and Discussion

In this paper, we extend concepts of the *birth* time and *death* time to the interaction for proposing a new measure that quantifies the interaction strength. These concepts are originally proposed for identifying true topology features (e.g., connected components and loops). Rigorously speaking, the structure of interest for interaction detection is a sub-network that connects a group of input features with output neurons, which is not a well-defined algebraic topology concept. Therefore, a lot of theoretical properties of the subject across the filtration is lost. However, to the best of our knowledge, by extending these concepts from Persistent Homology, we propose the first NN specific interaction strength measure with stability guarantee (Theorem 1). Furthermore, we derived a topology-motivated algorithm to compute the interaction strength efficiently (Lemma 1).

We note that as NNs contain only 1-simplex, many of these topological properties degenerate to the field of graph theory. The proposed filtration process is equivalent to building maximum spanning trees (MSTs) of NNs using the Kruskal algorithm. The proposed persistence of feature groups is the gap length between MSTs of two sub-networks. It would be interesting to consider the theoretical benefit of our proposed measure from the perspective of graph theory. We leave it as future work.

Also, we want to emphasize that our image experiment in section 4.3 is exploratory. This experiment is designed to illustrate that the proposed PID is capable of detecting extreme high-order interactions in a specific input. Moreover, the saliency map obtained by utilizing PID could also provide visual cues for understanding how CNNs make decisions. We note that PID is complementary to most existing explainable-CV works. Especially, the saliency map in section 4.3 is obtained only from the interaction effects between raw pixels. In contrast, most explainable-CV works (e.g., Grad-CAM [30]) only consider how a specific raw-pixel influence the decision of models, and the interaction effects are ignored by them because most of these works do not access to Hessian Matrix or compute the approximation of Hessian Matrix.

## 6    Conclusion

In this work, we propose a theoretically well-defined measure for quantifying interaction strength by investigating the topology of neural networks. We show that this measure captures topological information that pertains to learned interactions in neural networks. Based on this measure, we derive the PID algorithm to detect interactions. We also give the theoretical analysis for it and show how to extend our method to local interaction detection. We demonstrate our proposed method has the practical utility of accurately detecting feature interactions without the need to prespecify interaction types or to search an exponential solution space of interaction candidates.

## Statement of Broader Impact

The proposed PID algorithm can be applied in various fields because it provides knowledge about a domain. Any researcher who needs to design experiments might benefit from our proposed algorithm in the sense that it can help researchers formulate hypotheses that could lead to new data collection and experiments. For example, PID can help us discover the combined effects of drugs on human body: By utilizing PID on patients' records, we might find using Phenelzine togther with Fluoxetine has a strong interaction effect towards serotonin syndrome. Thus, PID has great potential in helping the development of new therapies for saving lives.

Also, this project will lead to effective and efficient algorithms for finding useful any-order crossing features in an automated way. Finding useful crossing features is one of the most crucial task in the Recommender Systems. Engineers and Scientists in E-commerce companies may benefit from our results that our algorithm can alleviate the human effect on finding these useful patterns in the data.

## Acknowledgements

We would like to sincerely thank everyone who has provided their generous feedback for this work. Thank the anonymous reviewers for their thorough comments and suggestions. The authors thank the Texas A&M College of Engineering and Texas A&M University.

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
