[Supplementary Material]

# Appendices

## A  The Persistence Interaction Detection Algorithm

---

**Algorithm 1:** The proposed Persistence Interaction Detection (PID) algorithm

---

**Input:** A trained feed-forward neural network, target layer $l$, norm $p$.

**Output:** ranked list of interaction candidates $\{\mathcal{I}_i\}$.

**1** Construct size pair $(\mathcal{G}, \phi)$ and its filtration $\mathcal{G}^{w'_0} \subseteq ... \subseteq \mathcal{G}^{w'_n}$

**2** $\mathcal{K} \leftarrow$ initialize an empty dictionary mapping interaction candidate to persistence

**3 for** $i$=0:n **do**

**4**    $\lambda \leftarrow w'_i; \quad \mathcal{G}^\lambda \leftarrow \mathcal{G}^{w'_i}$

**5**    Calculate $\mathbf{M}^{\lambda_{up}}_{(l)}$ and $\mathbf{M}^{\lambda_{down}}_{(l)}$ according to Equation (3)

**6**    **for** each row $\mathbf{m}$ of $\mathbf{M}^{\lambda_{down}}_{(l)}$ indexed by $r$ **do**

**7**       **if** all elements in $r^{\text{th}}$ column of $\mathbf{M}^{\lambda_{up}}_{(l)}$ are 0 **then**

**8**          **continue** // $r$-th unit in $l$-th layer is not connected with any
               final output units

**9**       **end**

**10**       $\mathcal{I} \leftarrow$ initialize an empty set;

**11**       **for** $j$=0:$d$-1 **do**

**12**          **if** $m_j == 0$ **then**

**13**             **continue** // $r$-th unit is not connected with feature $j$

**14**          **end**

**15**          $d_\mathcal{I} \leftarrow \lambda$ // $\mathcal{I}$ merged with $j$

**16**          ;

**17**          $b_{\mathcal{I} \cup j} \leftarrow \lambda$;

**18**          $\mathcal{K}[\mathcal{I}] \leftarrow \mathcal{K}[\mathcal{I}] + |b_\mathcal{I} - d_\mathcal{I}|^p$;

**19**          $\mathcal{I} \leftarrow \mathcal{I} \cup j$ ;

**20**       **end**

**21**    **end**

**22 end**

**23** $\{\mathcal{I}_i\} \leftarrow$ interaction candidates in $\mathcal{K}$ sorted by their strengths in descending order.

---

Our PID framework is presented in Algorithm 1. Besides the $\langle \phi = \lambda \rangle$-connectivity between $\mathcal{I}$ and final outputs, we also consider: First, whether $\mathcal{I}$ and a particular neuron $r$ are connected; second, whether the neuron $r$ and final outputs are connected under the threshold $\lambda$. Recall that the measuring function $\phi$ is non-decreasing over $\mathcal{G}$ (Definition 2), and the birth time and death time of each interaction candidates can be determined through one pass of all thresholds. As shown in Figure 3, calculating the interaction strength of $\mathcal{I}$ at neuron $r$ is equivalent to running Algorithm 1 on a neural network whose $l^{\text{th}}$ layer is only composed by neuron $r$.

The time complexity of PID is $\mathcal{O}(Ndp_l)$, where $N$ denotes the total number of weights used as thresholds in the filtration and $p_l$ is the number of neurons at target layer $l$. One possible way to reduce the time complexity is that, we can change the $\mathcal{W}' := \{|w|/w_{max}|w \in \mathcal{W}\}$ in section 2.2 to $\mathcal{W}' := \{|w|/w_{max}|w \in \mathcal{W} \wedge w \geq \eta w_{max}\}$, where $\eta$ is a hyperparameter which controls total number of weights used as thresholds in Algorithm 1. We do not utilize this method to accelerate PID in all experiments of this paper (i.e., set $\eta$ as 0).

## B  Proof of Lemma 1

**Lemma 1** (Proof in Appendix B). *Let $\{0, ..., d-1\}$ denotes the input feature set, and $\mathbf{M}^\lambda$ denotes the aggregated mask matrix corresponding to threshold $\lambda$, where the $r^{\text{th}}$ row of $\mathbf{M}^\lambda$ is denoted as $\mathbf{m}^\lambda_r \in \mathbb{R}^d$. The feature subset $\mathcal{I}$ and the corresponding $r^{th}$ unit at the final output layer are $\langle \phi = \lambda \rangle$-connected if all elements in $[\mathbf{m}^\lambda_r]^\mathcal{I} \in \mathbb{R}^{|\mathcal{I}|}$ are non-zero and all other elements in $[\mathbf{m}^\lambda_r]^{\{0,...,d-1\}\backslash\mathcal{I}}$ are zero, where $[\mathbf{m}^\lambda_r]^\mathcal{I}$ is the subvector of $\mathbf{m}^\lambda_r$ selected by $\mathcal{I}$.*

We obtain Lemma 1 following from the theoretical analysis in Appendix E of [7].

*Proof.* If the network has exactly one layer, $\mathbf{M}^\lambda = (\mathbf{M}^\lambda_{(1)})^\top$ directly gives the connectivity between input features and output units in the final output layer.

In cases when $\mathbf{M}^\lambda$ has more than one hidden layer, first consider the weight connectivity between input features and the second hidden layer. Since a feed-forward neural network is a directed acyclic graph and a hop is a transition from one layer to the next, we can view the connectivity from input features to the second hidden layer as two hops or two applications of an adjacency matrix, $\mathbf{A}$, comprising of $\mathbf{M}^\lambda_{(2)}$ and $\mathbf{M}^\lambda_{(1)}$ as:

$$\mathbf{A} = \begin{bmatrix} 0 & (\mathbf{M}^\lambda_{(1)})^\top & 0 \\ 0 & 0 & (\mathbf{M}^\lambda_{(2)})^\top \\ 0 & 0 & 0 \end{bmatrix}.$$

Therefore, the adjacency matrix for two hops is:

$$\mathbf{A}^2 = \begin{bmatrix} 0 & 0 & (\mathbf{M}^\lambda_{(2)})^\top (\mathbf{M}^\lambda_{(1)})^\top \\ 0 & 0 & 0 \\ 0 & 0 & 0 \end{bmatrix}.$$

Since the elements of $\mathbf{A}^2$ are the number of paths between graph vertices in two hops, the non-zero elements of $(\mathbf{M}^\lambda_{(2)})^\top (\mathbf{M}^\lambda_{(1)})^\top$ represent the existence of paths from features to the second hidden layer, and the zero elements represent the lack of such paths. We can therefore repeatedly apply hops up to the $L^{th}$ hidden layer, yielding $(\mathbf{M}^\lambda_{(L)})^\top \cdot (\mathbf{M}^\lambda_{(L-1)})^\top \cdots (\mathbf{M}^\lambda_{(1)})^\top$ to represent the zero and non-zero paths from input features to the neurons in the $L^{th}$ layer. Thus, if all elements in $[\mathbf{m}^\lambda_r]^\mathcal{I}$ are non-zero and all other elements in $[\mathbf{m}^\lambda_r]^{\{0,\dots,d-1\}\backslash\mathcal{I}}$ are zero, $\mathcal{I}$ and unit $r$ are $\langle \phi = \lambda \rangle - connected$ by Definition 3.

□

## C   Proof of Theorem 1

In this subsection, we will prove Theorem 1 and evaluate it empirically. We first give the stability lemma for connected components and then utilize it to derive Theorem 1.

**Definition 4** (Hausdorff distance). *For points $p = (p_1, p_2)$ and $q = (q_1, q_2)$ in $\mathbb{R}^2$, let $\|p - q\|_\infty$ be the maximum of $|p_1 - q_1|$ and $|p_2 - q_2|$. Let $\|f - g\|_\infty = \sup_x |f(x) - g(x)|$. Let $X$ and $Y$ be multisets of points. The Hausdorff distance is defined as*

$$d_H(X, Y) = \max\{\sup_{x \in X} \inf_{y \in Y} \|x - y\|_\infty, \sup_{y \in Y} \inf_{x \in X} \|x - y\|_\infty\},$$

For two feed forward neural networks $f$ and $g$ with the exact same architecture, let $g$ be a neural network that is obtained by perturbing the weights of $f$. The corresponding size pairs $(\mathcal{G}_f, \phi_f)$ and $(\mathcal{G}_g, \phi_g)$ are constructed following instructions in Section 2.2. Let $\delta = \max_{e \in E} |\phi_f(e) - \phi_g(e)|$ be the magnitude of the perturbation, i.e., $\|\phi_f - \phi_g\|_\infty = \delta$. Persistence diagrams of $(\mathcal{G}_f, \phi_f)$ and $(\mathcal{G}_g, \phi_g)$ are denoted as $\mathcal{D}[(\mathcal{G}_f, \phi_f)]$ and $\mathcal{D}[(\mathcal{G}_g, \phi_g)]$, respectively. We note that $\phi_f$ and $\phi_g$ are piecewise linear functions on simplicial complexes, where a simplicial complex is a high-dimensional generalization of a graph in topological space. Piecewise linear functions satisfy the following Lemma:

**Lemma 2** (Proof in [31]). $d_H(\mathcal{D}[(\mathcal{G}_f, \phi_f)], \mathcal{D}[(\mathcal{G}_g, \phi_g)]) \leq \delta$.

When weights in the networks are perturbed, the birth time and death time of connected components are also changed. Lemma 2 shows that the Hausdorff distance between the persistence diagrams is bounded by the magnitude of the perturbation, i.e., for the set of all connected components $\mathcal{J}$, suppose its birth time $b_{\mathcal{J}}$ and death time $d_{\mathcal{J}}$ changes to $b'_{\mathcal{J}}$ and $d'_{\mathcal{J}}$, then $\max(|b_{\mathcal{J}} - b'_{\mathcal{J}}|, |d_{\mathcal{J}} - d'_{\mathcal{J}}|) \leq \delta$.

For any interaction candidate $\mathcal{I}$ that are detected in both $f$ and $g$ by Algorithm 1, we denote the birth time of $\mathcal{I}$ in $f$ and $g$ as $b_{\mathcal{I}}$ and $b'_{\mathcal{I}}$, respectively. Similarly, we use $d_{\mathcal{I}}$ and $d'_{\mathcal{I}}$ for the death time of

$\mathcal{I}$ in $f$ and $g$, respectively. Suppose the connected component $\mathcal{J}$ and the connected component $\mathcal{J}'$ cause the birth of interaction $\mathcal{I}$ in $f$ and $g$, respectively. We have the following corollary:

**Corollary 1.** $|b_\mathcal{I} - b'_\mathcal{I}| \le 3\delta.$

*Proof.* From Definition 3, we have $|b_\mathcal{I} - b'_\mathcal{I}| = |b_\mathcal{J} - b_{\mathcal{J}'}| = |\min_{e \in \mathcal{J}} \phi_f(e) - \min_{e \in \mathcal{J}'} \phi_g(e)|$. If $\mathcal{J}$ and $\mathcal{J}'$ are composed of identical set of edges, then we can directly prove Corollary 1 following Lemma 2. If $\mathcal{J}$ and $\mathcal{J}'$ contain different edges, without loss of generality, let $\mathcal{J}' \backslash \mathcal{J} \ne \emptyset$. $\forall e, \phi_f(e) \le \min_{e' \in \mathcal{J}} \phi_f(e') - 2\delta$, we have the following inequality:

$$\phi_g(e) \le \min_{e' \in \mathcal{J}} \phi_g(e'). \tag{4}$$

Inequality (4) follows from the fact that $\forall e, |\phi_f(e) - \phi_g(e)| \le \delta$, which implies that, $\forall e, \phi_f(e) \le \min_{e' \in \mathcal{J}} \phi_f(e') - 2\delta$, $e$ has to wait for all edges in $\mathcal{J}$ to be added to the filtration before being added itself. Namely, $\mathcal{I}$ is born before the threshold arrives at $\phi_g(e)$ and, consequently, $e \notin \mathcal{J}'$. Thus, $\forall e, e \in \mathcal{J}' \backslash \mathcal{J}$, $e$ satisfies $\phi_f(e) \ge \min_{e' \in \mathcal{J}} \phi_f(e') - 2\delta$. Following this fact, we have

$$
\begin{aligned}
\min_{e \in \mathcal{J}'} \phi_g(e) &= \min\{\min_{e \in \mathcal{J}' \cup \mathcal{J}} \phi_g(e), \min_{e \in \mathcal{J}' \backslash \mathcal{J}} \phi_g(e)\} \\
&\ge \min\{\min_{e \in \mathcal{J}' \cup \mathcal{J}} \phi_f(e) - \delta, \min_{e \in \mathcal{J}' \backslash \mathcal{J}} \phi_f(e) - \delta\} \tag{5} \\
&\ge \min\{\min_{e \in \mathcal{J}' \cup \mathcal{J}} \phi_f(e) - \delta, \min_{e \in \mathcal{J}} \phi_f(e) - 3\delta\} \tag{6} \\
&\ge \min\{\min_{e \in \mathcal{J}} \phi_f(e) - \delta, \min_{e \in \mathcal{J}} \phi_f(e) - 3\delta\} \\
&= \min_{e \in \mathcal{J}} \phi_f(e) - 3\delta. \tag{7}
\end{aligned}
$$

Inequality (5) follows from $\forall e \in E, |\phi_f(e) - \phi_g(e)| \le \delta$. The inequality (6) follows from the fact that $\forall e \in \mathcal{J}' \backslash \mathcal{J}, \phi_f(e) \ge \min_{e' \in \mathcal{J}} \phi_f(e') - 2\delta$. By equation (7), we have $b_\mathcal{J} - b_{\mathcal{J}'} \le 3\delta$.

By exchanging $f$ with $g$, we have $b_{\mathcal{J}'} - b_\mathcal{J} \le 3\delta$. Combining them together finishes the proof.

$\square$

It is trivial to show that Corollary 1 can be extended to the death time, i.e., we also have $|d_\mathcal{I} - d'_\mathcal{I}| \le 3\delta$.

After proving Corollary 1, we return to prove the theorem.

**Theorem 1** (Proof and empirical analysis in Appendix C). *Let $\delta = \max_{e \in E} |\phi_f(e) - \phi_g(e)|$ be the magnitude of perturbation. For all interaction candidate $\mathcal{I}$ that are both detected in $f$ and $g$ by Algorithm 1, it satisfies $|\rho_f(\mathcal{I}) - \rho_g(\mathcal{I})| \le C\delta$.*

*Proof.* In Algorithm 1, interaction candidates are generated at each neuron $r$ of a particular layer $l$. As shown in Figure 3, calculating the interaction strength of $\mathcal{I}$ at neuron $r$ is equivalent to running Algorithm 1 on a neural network whose $l^{\text{th}}$ layer is only composed by neuron $r$. Thus Corollary 1 also holds for interaction candidate $\mathcal{I}$ generated at each neuron. We use $\text{per}_f^{(r)}(\mathcal{I})$, $b^{(r)}(\mathcal{I})$, and $d^{(r)}(\mathcal{I})$ to represent the persistence, the birth time, and the death time of $\mathcal{I}$ generated at neuron $r$ corresponding to $f$, respectively. Similarly, for $g$, the persistence, the birth time, and the death time of $\mathcal{I}$ generated at neuron $r$ are denoted as $\text{per}_g^{(r)}(\mathcal{I})$, $b'^{(r)}(\mathcal{I})$, and $d'^{(r)}(\mathcal{I})$, respectively.

$$
\begin{aligned}
\text{per}_f^{(r)}(\mathcal{I}) &= |b_\mathcal{I}^{(r)} - d_\mathcal{I}^{(r)}| \\
&= |b_\mathcal{I}^{(r)} - b_\mathcal{I}'^{(r)} + b_\mathcal{I}'^{(r)} - d_\mathcal{I}'^{(r)} + d_\mathcal{I}'^{(r)} - d_\mathcal{I}^{(r)}| \\
&\le \text{per}_g^{(r)}(\mathcal{I}) + 6\delta,
\end{aligned}
$$

By exchanging $f$ with $g$, we have $\text{per}_g^{(r)}(\mathcal{I}) \leq 6\delta + \text{per}_f^{(r)}(\mathcal{I})$. Combining them together, we have $|\text{per}_f^{(r)}(\mathcal{I}) - \text{per}_g^{(r)}(\mathcal{I})| \leq 6\delta$. Then it follows

$$
\begin{aligned}
|\rho_f(\mathcal{I}) - \rho_g(\mathcal{I})| &= |\sum_{r \in l^{th}\text{layer}} [\text{per}_f^{(r)}(\mathcal{I})]^p - [\text{per}_g^{(r)}(\mathcal{I})]^p| \\
&\leq p|\sum_{r \in l^{th}\text{layer}} [\text{per}_f^{(r)}(\mathcal{I}) - \text{per}_g^{(r)}(\mathcal{I})] \max\{\text{per}_f^{(r)}(\mathcal{I}), \text{per}_f^{(r)}(\mathcal{I})\}^{p-1}| \\
&\leq 6pN_l\delta.
\end{aligned} \tag{8}
$$

Where $N_l$ is the number of units in layer $l$. The inequality (8) follows from the fact that $\max\{\text{per}_f^{(r)}(\mathcal{I}), \text{per}_f^{(r)}(\mathcal{I})\} \leq 1$.

$\square$

Beyond Theorem 1, there exists the corner case that there are interaction candidates only detected in one neural network, but not the other. We will show that this corner case only happens if $\delta$ is greater than a threshold.

Let $[d] := \{0, \cdots, d-1\}$ be the input feature set. Without loss of generality, suppose interaction candidate $\mathcal{I} \subset [d]$ only born in $f$, but not in $g$; and the connected component $\mathcal{J}$ cause the birth of $\mathcal{I}$ in $f$. Let $g$ be a neural network that is obtained by perturbing the weights of $f$. According to Definition 3, if $\mathcal{I}$ only born in $f$, it means that there exists some edges corresponding to the connection between input features and hidden units in the first layer, which satisfy the following:

$$
\exists e' \in [d]\backslash\mathcal{I},
$$
$$
s.t. \quad \phi_f(e') \geq \min_{e \in \mathcal{J}} \phi_f(e) - 2\delta \tag{9}
$$

The above inequality follows from the fact that $\forall e, |\phi_f(e) - \phi_g(e)| \leq \delta$, which implies that, $\forall e, \phi_f(e) \leq \min_{e' \in \mathcal{J}} \phi_f(e') - 2\delta$, $e$ has to wait for all edges in $\mathcal{J}$ to be added to the filtration before being added itself. Therefore, if $\mathcal{I}$ does not born in $g$, there must $\exists e' \in [d]\backslash\mathcal{I}$ such that $\phi_f(e') \geq \min_{e \in \mathcal{J}} \phi_f(e) - 2\delta$. In conclusion, if $\mathcal{I}$ only detected in $f$, the perturbation magnitude $\delta$ must satisfy:

$$
\delta \geq \frac{\min_{e \in \mathcal{J}} \phi_f(e) - \max_{e \in [d]\backslash\mathcal{I}} \phi_f(e)}{2} \tag{10}
$$

Table 3: Perturbation results.

| $\delta$ | $|\rho_f(\mathcal{I}) - \rho_g(\mathcal{I})|$ |
|---|---|
| 0.001 | 0.0935 |
| 0.01 | 0.1645 |
| 0.1 | 0.1990 |
| 1 | 0.3321 |

Here we randomly perturb the weights of an MLP trained on synthetic dataset $F_1$, which has architecture of 64-32-16 first-to-last hidden layer sizes. The layer $l$ in Algorithm 1 is set to the first layer, and the norm $p$ in Algorithm 1 is set to 2. The results are shown in Table 3.

## D  Extensibility

In this section, first, we show how to extend PID to CNNs. Second, we introduce how to extend our method to local interaction detection.

Let $\mathbf{H} \in \mathbb{R}^{height \times width}$ be a convolution kernel and $\mathbf{X} \in \mathbb{R}^{H \times W}$ be a tensor. Let $*$ refer to the convolution operation. Suppose the height and width of the $\mathbf{H} * \mathbf{X}$ are $H_{\text{out}}$ and $W_{\text{out}}$, respectively. We define $\mathcal{H} \in \mathbb{R}^{H_{\text{out}} \times W_{\text{out}} \times H \times W}$ as the corresponding four dimensional convolution tensor such that:

$$
\mathcal{H}(i, j, i : i + height, j : j + width) = \mathbf{H},
$$

for $\forall i \in [0, H_{\text{out}}), \forall j \in [0, W_{\text{out}})$. Then we have the following equation:

$$\mathbf{H} * \mathbf{X} = \mathcal{H} \otimes \mathbf{X}, \tag{11}$$

where $\mathcal{H} \otimes \mathbf{X}$ is the tensor product such that $[\mathcal{H} \otimes \mathbf{X}]_{i,j} = \sum_{k=0}^{H} \sum_{l=0}^{W} \mathcal{H}_{i,j,k,l} X_{k,l}$. Generally, for $\mathbf{H}' \in \mathbb{R}^{C_{\text{out}} \times C_{\text{in}} \times height \times width}$ and $\mathbf{X}' \in \mathbb{R}^{C_{\text{in}} \times H \times W}$, we can convert the convolution between $\mathbf{H}'$ and $\mathbf{X}'$ into $\mathbf{H}' * \mathbf{X}' = \mathcal{H}''\mathbf{X}''$ [32] using equation (11), where $\mathcal{H}'' \in \mathbb{R}^{C_{\text{out}} H_{\text{out}} W_{\text{out}} \times C_{\text{in}} HW}$ is the flattened matrix of $\mathcal{H}'$ with multi-channels, and $\mathbf{X}'' \in \mathbb{R}^{C_{in} HW}$ is the flattened vector of $\mathbf{X}'$. Then we can build the filtration of CNNs just as MLPs.

Given a data point $\mathbf{x} \in \mathbb{R}^d$, the feed forward neural network $f$ with ReLU activation function is a linear model in a region surrounding $\mathbf{x}$:

$$f(\mathbf{x}) = \mathbf{W}_{\mathbf{x}}^{(L)\top} \cdots \mathbf{W}_{\mathbf{x}}^{(1)\top} \mathbf{x}, \tag{12}$$

where $\mathbf{W}_{\mathbf{x}}^{(L)}$ is the equivalent weight matrix which combines the resulted activation pattern with $\mathbf{W}^{(L)}$, e.g., $\text{ReLU}(\mathbf{W}^{(1)\top}\mathbf{x}) = \mathbf{W}_{\mathbf{x}}^{(1)\top}\mathbf{x}$, where $\mathbf{W}_{\mathbf{x}}^{(1)}$ is modified from $\mathbf{W}^{(1)}$ by setting the columns, whose corresponding activation patterns are 0, to be all zero vectors. We denote the output value of the $i^{th}$ neuron in the $l^{th}$ layer, before activation, as $z_i^l$. From equation (12), for local interaction detection, the measuring function $\phi$ can be revised to

$$\phi((v_{l-1,i}, v_{l,j})) = \frac{|W_{i,j}^{(l)}|\text{ReLU}(z_i^{l-1})}{\Phi}, \tag{13}$$

where $\Phi = \max_{i,j,l} |W_{i,j}^{(l)}|\text{ReLU}(z_i^{l-1})$.

# E    Supplemental Material for the Synthetic Data Experiments

## E.1    Experiment Setting

Table 4: Test suite of data-generating functions.

| | |
|---|---|
| $F_1(x)$ | $\pi^{x_0 x_1} \sqrt{2x_2} - \sin^{-1}(x_3) + \log(x_2 + x_4) - \frac{x_8}{x_9}\sqrt{\frac{x_6}{x_7}} - x_1 x_6$ |
| $F_2(x)$ | $\pi^{x_0 x_1} \sqrt{2|x_2|} - \sin^{-1}(0.5x_3) + \log(|x_2 + x_4| + 1) + \frac{x_8}{1+|x_9|}\sqrt{\frac{x_6}{1+|x_7|}} - x_1 x_6$ |
| $F_3(x)$ | $e^{|x_0 - x_1|} + |x_1 x_2| - x_2^{2|x_3|} + \log(x_3^2 + x_4^2 + x_6^2 + x_7^2) + x_8 + \frac{1}{1+x_9^2}$ |
| $F_4(x)$ | $e^{|x_0 - x_1|} + |x_1 x_2| - x_2^{2|x_3|} + \log(x_3^2 + x_4^2 + x_6^2 + x_7^2) + x_8 + \frac{1}{1+x_9^2} + x_0^2 x_3^2$ |
| $F_5(x)$ | $\frac{1}{1+x_0^2+x_1^2+x_2^2} + \sqrt{e^{x_3+x_4}} + |x_5 + x_6| + x_7 x_8 x_9$ |
| $F_6(x)$ | $e^{|x_0 x_1 + 1|} - e^{|x_2 + x_3| + 1} + \cos(x_4 + x_5 - x_7) + \sqrt{x_7^2 + x_8^2 + x_9^2}$ |
| $F_7(x)$ | $(\tan^{-1}x_0 + \tan^{-1}x_1)^2 + \max(x_2 x_3 + x_5, 0) - \frac{1}{1+(x_3 x_4 x_5 x_6 x_7)^2} + (\frac{|x_6|}{1+|x_8|})^5 + \sum_{i=0}^9 x_i$ |
| $F_8(x)$ | $x_0 x_1 + 2^{x_2 + x_4 + x_5} + 2^{x_2 + x_3 + x_4 + x_6} + \sin(x_6 \sin(x_7 + x_8)) + \cos^{-1}(0.9 x_9)$ |
| $F_9(x)$ | $\tanh(x_0 x_1 + x_2 x_3)\sqrt{|x_4|} + e^{x_4 + x_5} + \log(x_5^2 x_6^2 x_7^2 + 1) + x_8 x_9 + \frac{1}{1+|x_9|}$ |
| $F_{10}(x)$ | $\sinh(x_1 + x_2) + \cos^{-1}(\tanh(x_2 + x_4 + x_6)) + \cos(x_3 + x_4) + \sec(x_6 x_8)$ |

The synthetic datasets in section 4.1 are shown in Table 4. $F_1$ is a commonly used function in interaction detection literature [3, 21, 33]. All features were uniformly distributed between -1 and 1 except in $F_1$, where we used the same variable ranges as those reported in [21]. In all synthetic experiments, we evenly split train set, validation set and test set on 30k data points. All networks consisted of four hidden layers with first-to-last layer sizes of: 140, 100, 60, and 20 units. All networks employed ReLU activation and were trained using Adam optimizer with a $5e-3$ learning rate cross all ten datasets. The L1 regularization strength was set to $5e-5$. The early stopping round was set to 100 to prevent overfitting. The mean square error of all trained MLPs are less than $3e-3$ on test data.

## E.2    Detailed Analysis

Main effects describe the univariate influences of features on outcomes [11], e.g., $\sin^{-1}(0.5x_3)$ in the synthetic dataset $F_2$. Main effects might entangle with true interactions, resulting in spurious interactions. For example, in $F_2$, $\{0, 1, 2\}$ is true interaction and $\{0, 1, 2, 3\}$ is a spurious interaction, which

Figure 5: Heat maps of pairwise interaction strengths proposed by our PID corresponding to Table 1. Cross-marks indicate ground truth interactions.

is an entanglement between true interaction $\{0, 1, 2\}$ and main effect $\{3\}$. Handling main effects is an important problem in interaction detection [4, 34, 35]. We remark that in synthetic experiments, higher AUCs indicates the interaction detection algorithms can more thoroughly disentangle main effects from true interactions.

In Figure 5, heat maps of synthetic functions show the relative strengths of all possible pairwise interactions proposed by PID, and the ground truth is indicated by red cross-marks. In general, the interaction strengths are higher at the cross-marks. Although most of the synthetic functions contain main effects, from Figure 5 and Table 1, the influence of main effects is limited: only the AUCs of $F_2$ and $F_7$ are under 0.9. We hypothesize that if a overparameterized neural network is trained with proper regularization, the neural network will push the modeling of main effect to a small portion of neurons at the first layer.

To confirm our hypothesis, here we analyze the MLP trained on synthetic dataset $F_3$. For $F_3$, main effects are $x_8$ and $\frac{1}{1+x_9^2}$. Let $\mathbf{W}^{(1)} \in \mathbb{R}^{d \times p_1}$ be the weight matrix of the first layer. The weights corresponding to input feature $r$ are the $r^{th}$ row of $\mathbf{W}^{(1)}$, which is denoted as $\mathbf{W}_{r,:}^{(1)}$. For convince, we mark different neurons at the first layer by their indices. In Figure 6, we show the statistics of magnitudes of $\mathbf{W}_{8,:}^{(1)}$ and $\mathbf{W}_{9,:}^{(1)}$ of an MLP trained on synthetic dataset $F_3$. In general, only a few neurons have large weights connecting to $x_8$ and $x_9$, which are corresponding to the peaks in Figure 6. We plot the weights of all input features to these neurons in Figure 7. To be specific, given a representative neuron $c$, we plot the weight statistics of input features to that neuron, which is denoted as $\mathbf{W}_{:,c}^{(1)}$. For $\mathbf{W}_{9,:}^{(1)}$, two peaks in Figure 6 have identical patterns. Here we only show statistics for one of them. For $\mathbf{W}_{8,:}^{(1)}$, we show weights statistics of all input features to neuron 36; For $\mathbf{W}_{9,:}^{(1)}$, we show weights statistics of all input features to neuron 53. This result is consistent with our hypothesis: neural networks will naturally separate different interactions in the first hidden layer.

Figure 6: Statistics of the magnitudes of $\mathbf{W}_{8,:}^{(1)}$ and $\mathbf{W}_{9,:}^{(1)}$ (the MLP is trained on $F_3$).

Figure 7: The magnitude of weights corresponding to different input features at the selected representative neurons in the first layer (these neurons are corresponding to the peaks in Figure 6).

## E.3 Sensitivity to the Architecture and Regularization Strength

We try to analyze the sensitivity of interaction detection algorithms to the architecture of MLPs. In Figure 8, 64 represents an MLP with first-to-last layer sizes of 64-32-16; 128 represents an MLP with the 128-64-32 architecture; 140 represents an MLP with the 140-100-60-20 architecture; and 256 represents an MLP with the 256-128-64 architecture. The training hyperparameters of these MLPs are identical to those reported in Appednix E.1. We ran ten trials of NID and PID on each dataset and removed two trials with the highest and the lowest AUC scores. The mean square errors of all MLP models used for detecting interactions are less than $3e-3$ on test data.

(a) Average AUCs of pairwise interaction detected by NID and PID using MLPs with different architectures.

(b) Average Mean Square Error (MSE) of MLPs with different architectures on test data.

Figure 8: The sensitivity analysis of interaction detection algorithms to the architecture of MLPs (L1 is set to $5e-5$).

(a) Average AUCs of pairwise interaction detected by NID and PID using MLPs with different architectures.

(b) Average Mean Square Error of MLPs with different architectures on test data.

Figure 9: The sensitivity analysis of interaction detection algorithms to the regularization strength (L1 is set to $5e-6$).

(a) Average AUCs of pairwise interaction detected by NID and PID using MLPs with different architectures.

(b) Average Mean Square Error of MLPs with different architectures on test data.

Figure 10: The sensitivity analysis of interaction detection algorithms to the regularization strength (L1 is set to $5e-4$).

Table 5: AUC of pairwise interaction strengths proposed by PID and NID on the synthetic functions. The L1 regularization strength is set to $5e-4$ here.

|  | NID | PID |
|---|---|---|
| $F_1(x)$ | $0.898 \pm 0.0145$ | $\mathbf{0.915 \pm 0.0144}$ |
| $F_2(x)$ | $0.700 \pm 0.0419$ | $\mathbf{0.717 \pm 0.0349}$ |
| $F_3(x)$ | $0.964 \pm 0.0318$ | $\mathbf{0.966 \pm 0.0342}$ |
| $F_4(x)$ | $0.928 \pm 0.0649$ | $\mathbf{0.938 \pm 0.0585}$ |
| $F_5(x)$ | $1.000 \pm 0.0000$ | $1.000 \pm 0.0000$ |
| $F_6(x)$ | $0.740 \pm 0.0531$ | $\mathbf{0.769 \pm 0.0669}$ |
| $F_7(x)$ | $\mathbf{0.807 \pm 0.0318}$ | $0.806 \pm 0.0385$ |
| $F_8(x)$ | $0.996 \pm 0.0085$ | $\mathbf{0.997 \pm 0.0084}$ |
| $F_9(x)$ | $0.785 \pm 0.0778$ | $\mathbf{0.811 \pm 0.0475}$ |
| $F_{10}(x)$ | $\mathbf{0.937 \pm 0.0285}$ | $0.927 \pm 0.0383$ |
| average | $0.876 \pm 0.1033$ | $\mathbf{0.885 \pm 0.0954}$ |

Figure 11: Heat maps of pairwise interaction strengths proposed by our PID corresponding to Table 1. Cross-marks indicate ground truth interactions. (L1 is set to $5e-4$).

The regularization strength controls the weight sparsity in neural networks. Intuitively, it significantly influences the interaction detection results because it will change the connectivity in networks. Here we change the L1 strength to $5e-4$ and $5e-6$, and all other experiment settings are identical.

Figure 9 shows the results using MLP with L1 set to $5e-6$. The average MSE of all MLP models used here is less than $2e-3$ on test data. Similar to Figure 8, PID can achieve better performance than NID but the gap is small. Figure 10 shows the results using MLP with L1 set to $5e-4$. The average MSE of all MLP models used here is less than $1e-2$ on test data. Comparing Figure 10

with Figure 8, the mean square error is worse but is acceptable. However, the false discovery rate increases dramatically. To better understand the impact of regularization strength, we further analyze the MLP of 140-100-60-20 architecture. Similar to Figure 5, we plot the heat map in Figure 11, and the detailed results are shown in Table 5. Comparing Table 5 with Table 1, both the performances of PID and NID dropped. Moreover, the AUCs of $F_6$ and $F_9$ dropped more than 0.1. Here we provide a detailed case study for MLPs trained on synthetic dataset $F_6$. Comparing Figure 11 with Figure 5, it should be noted that the interaction strength between $\{x_7, x_8, x_9\}$ is very small (near 0 actually). As [6] Appendix I points out, in synthetic dataset $F_6$, $\{x_7, x_8, x_9\}$ can be approximated as

$$\sqrt{x_7^2 + x_8^2 + x_9^2} \approx c + x_7^2 + x_8^2 + x_9^2.$$

In [6], the authors show that $\{x_7, x_8, x_9\}$ are modeled as spurious main effects in the MLP-M (the MLP-M is an MLP with optional univariate networks, which details can be found in [6] Figure 2). Here we hypothesize that, under strong regularization strength, they are also modeled as spurious main effects in MLPs. Figure 12 shows the weight statistics of the magnitudes of $\mathbf{W}_{7,:}^{(1)}$, $\mathbf{W}_{8,:}^{(1)}$, and $\mathbf{W}_{9,:}^{(1)}$ of an MLP trained on $F_6$. There is a similar pattern between Figure 12 and Figure 6. Similar to Figure 7, we further plot the weights of input features to the representative neurons corresponding to the peaks in Figure 13. We remark that, for all these neurons corresponding to peaks in Figure 12, they share a similar pattern. Therefore, we show only one of their statistics for illustrative purposes. In Figure 13, we select neuron 5, 131, and 19 for $\mathbf{W}_{7,:}^{(1)}$, $\mathbf{W}_{8,:}^{(1)}$, and $\mathbf{W}_{9,:}^{(1)}$, respectively. From Figure 13, it can be seen that MLP do not model the interaction $\{x_7, x_8, x_9\}$. Instead, they are modeled as spurious main effects.

Figure 12: Statistics of the magnitudes of weights corresponding to $x_7$, $x_8$ and $x_9$ at different neurons of the first layer (the MLP is trained on $F_6$ with L1 regularization strength set to $5e - 4$).

Figure 13: The magnitude of weights corresponding to different input features at the selected representative neurons of $x_7$, $x_8$ and $x_9$ (L1 is set to $5e - 4$).

In conclusion, both NID and PID are insensitive to the architecture of MLPs and both of them are sensitive to the regularization strength. A detailed case study for the impact of regularization strength is shown in Figure 11, Figure 12, and Figure 13. This suggests that we should carefully choose the regularization strength. From Figure 8, Figure 10, and Figure 9, PID always achieves better performance. Also, we observe PID is more resilient to changes in regularization strength. Generally speaking, interaction detection algorithms have better AUC when the MLP has better performance. It makes sense that, when the MLP fits the true distribution, the interactions encoded in the networks are more accurate.

# F  Details for the Automatic Feature Engineering Experiments

## F.1  Experiment Setting

Table 6: Statistics of datasets. "# Dense"and "# Sparse" are the number of numerical features and the number of categorical features, respectively. "# Samples" is total available samples in each dataset.

| Dataset | #Samples | # Features | |
|---|---|---|---|
| | | # Dense | # Sparse |
| Amazon Employee | 32769 | 0 | 9 |
| Higgs Boson | 98050 | 28 | 0 |
| Creditcard | 284807 | 30 | 0 |
| Spambase | 4601 | 57 | 0 |
| Diabetes | 768 | 8 | 0 |

We perform most of the experiments on five open-source tabular datasets from different domains: **Amazon Employee**[2], **Higgs Boson**[3], **Creditcard** [4], **Spambase**[5] and **Diabetes**[6]. For the ease to reproduce our results, we use OpenML [36] to obtain all these datasets and adopt standard cross validation provided by OpenML. The statistics of datasets we used in Section 4.2 is described in Table 6.

The MLPs for NID and PID have architectures of 256-128-64 first-to-last hidden layer sizes, and they are trained with learning rate of $5e - 3$, batchsize of 100, and the Adam optimizer. As pointed out in Appendix E.3, the regularization strength significantly influences the results of NID and PID. We tune the L1 regularization strength with a search space $[1e - 6, 1e - 1]$ for each dataset. The early stopping round is set to 20 to prevent overfitting.

The synthetic feature $\mathbf{x}_{\mathcal{L}_i}$ is created by explicitly crossing sparse features indexed in $\mathcal{L}_i$. If interaction $\mathcal{L}_i$ involves dense features, we bucketize the dense features before crossing them. The bucket size is set to 100 across all experiments. Let $|\mathcal{L}_i| = t$ and $\{0, ..., t - 1\}$ is the interaction candidate specified by $\mathcal{L}_i$. A synthetic feature $\mathbf{x}_{\mathcal{L}_i}$ is an $t$-ary Cartesian product among $t$ features, which means $\mathbf{x}_{\mathcal{L}_i}$ takes on all possible values in $\{(x_1, ..., x_t) | \forall x_i \in \mathbf{x}_i, i = 0, ..., t - 1\}$.

Concerning the cardinality of synthetic features can be extremely large, yet many combinations do not exist in the training data, we limit the order of crossing features up to 4 over all five datasets. For sparse categorical features, like CatBoost [37], we apply target encoding to make them applicable to the random forest.

We run five trials of PID and NID on each dataset to obtain five different sets of top ten interactions. For each set of top ten interactions, we construct synthetic features and integrate them with original input features, and then we split the concatenated data into five folds. Subsequently, five random forest models are trained and evaluated with each fold given a chance to be the test set. Totally, we trained 25 random forest models on each dataset and removed two models with the highest and the lowest performance. We implement the random forest via LightGBM [38]. The hyperparameters of random forest is summerized in Table 7.

## F.2  Additional Experiment Results

The statistics of detected interaction orders by PID and NID are shown in Table 8. Interaction orders are averaged over 5 folds of cross-validation.

Here we present the case study for the "Amazon Employee" dataset in Table 9 and Table 10. The main reasons for choosing "Amazon Employee" are as follows: first, it is a dataset used for Kaggle challenges and, thus, the top solution is available. Second, the key technique in the top solution is to construct synthetic features for 2-order and 3-order interactions, so we can compare our detected interactions against the best hand-crafted interactions.

Table 7: Hyperparameters of the random forest.

| Name | Value |
|------|-------|
| early_stopping_rounds | 50 |
| num_boost_round | 5000 |
| learning_rate | 0.05 |
| lambda_1 | 0.2 |
| lambda_2 | 0.2 |
| bagging_raction | 0.85 |
| bagging_req | 3 |

Table 8: Interaction order statistics.

| Method | | Amazon Employee | Higgs Boson | Creditcard | Spambase | Diabetes |
|--------|------|-----------------|-------------|------------|----------|----------|
| | max | $4.00 \pm 0.00$ | $4.00 \pm 0.00$ | $4.00 \pm 0.00$ | $4.00 \pm 0.00$ | $3.60 \pm 0.80$ |
| NID | mean | $3.30 \pm 0.06$ | $2.50 \pm 0.11$ | $2.70 \pm 0.17$ | $2.62 \pm 0.12$ | $2.30 \pm 0.17$ |
| | min | $2.00 \pm 0.00$ | $2.00 \pm 0.00$ | $2.00 \pm 0.00$ | $2.00 \pm 0.00$ | $2.00 \pm 0.00$ |
| | max | $4.00 \pm 0.00$ | $3.80 \pm 0.40$ | $4.00 \pm 0.00$ | $4.00 \pm 0.00$ | $4.00 \pm 0.00$ |
| PID | mean | $3.30 \pm 0.14$ | $2.64 \pm 0.20$ | $2.84 \pm 0.31$ | $2.94 \pm 0.24$ | $3.48 \pm 0.35$ |
| | min | $2.00 \pm 0.00$ | $2.00 \pm 0.00$ | $2.00 \pm 0.00$ | $2.00 \pm 0.00$ | $2.40 \pm 0.49$ |

Table 9: Top ten interaction candidates proposed by PID for Amazom Employee dataset.

| Interaction Candidates | Interaction Strength |
|------------------------|----------------------|
| {RESOURCE, MGR_ID, ROLE_FAMILY_DESC} | 2.206 |
| {RESOURCE, MGR_ID} | 1.456 |
| {RESOURCE, MGR_ID, ROLE_DEPTNAME, ROLE_FAMILY_DESC} | 1.333 |
| {MGR_ID, ROLE_FAMILY_DESC} | 0.418 |
| {RESOURCE, MGR_ID, ROLE_DEPTNAME} | 0.393 |
| {RESOURCE, MGR_ID, ROLE_TITLE, ROLE_FAMILY} | 0.385 |
| {RESOURCE, MGR_ID, ROLE_ROLLUP_2, ROLE_FAMILY_DESC} | 0.315 |
| {RESOURCE, MGR_ID, ROLE_TITLE, ROLE_FAMILY_DESC} | 0.270 |
| {RESOURCE, MGR_ID, ROLE_FAMILY} | 0.220 |
| {MGR_ID, ROLE_DEPTNAME} | 0.190 |

Table 10: Top ten interaction candidates proposed by NID for Amazom Employee dataset.

| Interaction Candidates | Interaction Strength |
|------------------------|----------------------|
| {RESOURCE, MGR_ID, ROLE_FAMILY_DESC} | 26.757 |
| {RESOURCE, MGR_ID} | 22.060 |
| {RESOURCE, MGR_ID, ROLE_DEPTNAME, ROLE_FAMILY_DESC} | 10.423 |
| {MGR_ID, ROLE_FAMILY_DESC} | 7.713 |
| {RESOURCE, MGR_ID, ROLE_TITLE, ROLE_FAMILY_DESC} | 2.697 |
| {RESOURCE, MGR_ID, ROLE_FAMILY_DESC, ROLE_CODE} | 2.448 |
| {RESOURCE, ROLE_FAMILY_DESC} | 2.436 |
| {RESOURCE, MGR_ID, ROLE_ROLLUP_2, ROLE_FAMILY_DESC} | 2.316 |
| {RESOURCE, MGR_ID, ROLE_FAMILY_DESC, ROLE_FAMILY} | 1.187 |
| {ROLE_CODE, MGR_ID, ROLE_TITLE, ROLE_FAMILY_DESC} | 1.070 |

In general, the interaction candidates detected by NID and PID are similar. However, there exists some interaction candidates only detected by PID or NID, respectively. For example, "{MGR_ID, ROLE_FAMILY_DESC}" are only detected by NID. We note that the scale of the interaction strength proposed by PID and NID are different and only the rankings of interaction candidates are comparable. From Table 9, most of the interaction candidates proposed by PID for Amazon Employee are 3-order interactions. None of the top ranked interactions contain the input feature ROLE_CODE. This result is consistent with the top solution: "Transform the data to higher degree features by considering all

pairs and triples of the original data ignoring `ROLE_CODE`"[7]. In contrast, "`ROLE_CODE`" are contained in the interaction candidates proposed by NID. And our top ranked interactions are also consistent with the hand-designed synthetic features built from interactions,[8] such as {`RESOURCE, MGR_ID`} corresponding to "The number of unique resources that a MGR_ID received requests for".

## G  Details for High-order Interaction Detection on Image Datasets

Figure 14: Saliency maps of interaction strength found from applying PID on the CNN trained on MNIST dataset.

The neural network is composed of two convolutional layers of kernel size 5 and stride 1, followed by a max pooling layer and ReLU activation, and ended with a dense layer. The two convolutional layers contain 8 and 16 filters, respectively. It is trained with learning rate of $5e - 3$, batchsize of 100, the Adam optimizer, L1 regularization of $5e - 4$, and train epochs of 5.

Similar to Figure 4, Figure 14 also shows that PID are capable of detecting high-order interactions that represent object shapes.

## Footnotes

[2]https://www.kaggle.com/c/amazon-employee-access-challenge

[3]https://archive.ics.uci.edu/ml/datasets/HIGGS

[4]https://www.openml.org/d/1597

[5]https://archive.ics.uci.edu/ml/datasets/spambase

[6]https://www.openml.org/d/37

[7]https://www.kaggle.com/c/amazon-employee-access-challenge/discussion/4838

[8]https://www.kaggle.com/c/amazon-employee-access-challenge/discussion/5283