[Reviews · NeurIPS 2020]

Review 1

Summary and Contributions: The authors consider the problem of detecting interactions between feature elements in the feature space learned by a neural network. With the observation that such interactions can be modeled after the actual network connectivity, the authors propose a method based on topological analysis to address this issue. This is achieved by means of a new quantitative metric called PID that measures the interaction between input and output units in a neural network. The authors provide both theoretical as well as experimental analyses to corroborate their main ideas. Please see some notes below.

Strengths: Strengths - The paper is very well written and it is quite easy to follow the key ideas, motivations, and considerations. One suggestion for the authors here would be to consider including an introductory teaser figure in the front that encompasses the problem and the key ideas- this will help with material exposition. - The idea of using topological analysis in determining connectivity and interaction strength by formulating the problem as one of measuring graph persistence is interesting and well-motivated.

Weaknesses: Weaknesses Evaluating feature interactions - I think the synthetic dataset experiment in Section 4.1 is a good step towards evaluating the efficacy of PID. Looking at the AUC numbers, the performance of PID, AG, and NID seems quite close. Given these are synthetic data results, and the testbed is quite controlled, I would encourage the authors to provide more insights on why the performance is similar/close. For instance, the authors note AG is "tree-based", but it is not immediately clear how or why this may the main reason PID to perform better in F5, F6, and F8. Furthermore, with NID being a similar (in spirit) baseline, I would expect more in-depth analysis and discussion on the benefits that PID brings comparatively. Interaction detection in images - I think we would need more information here to understand the problem setup. Are the CNNs trained to perform categorical prediction, i.e., classify images? This does seem to be the case but I'd request the authors to clarify. If this is the case, when Q4 talks about understanding the behavior of CNNs, my understanding is this refers to identifying which regions in the input image are critical for the trained CNN to classify that particular image as belonging to a certain category. In the explainable CV literature, there is much recent work, with GradCAM being a notable example, that starts from the output prediction and uses various ways (e.g., gradients) to determine pixel regions that are important, producing saliency maps similar to Fig 4. Since the authors mention Q4 is fairly broad terms, I would expect discussion and comparison w.r.t. existing methods in this domain (at least CAM or GradCAM) - Additionally, I do not think results on MNIST and FashionMNIST datasets are enough to convincingly note PID can help explain behavior of CNNs. I encourage the authors to evaluate on more challenging classification scenariors, including at least ImageNet. Finally, to better understand the broader impact of PID in the context of CNNs, one would have to look at objectives other than classification, e.g., regression, metric learning, etc. While I do not expect results with these tasks already, I would encourage the authors to at least discuss how PID can be used in these cases. Minor, typos etc L38, L145: homolgy-> homology L94: measureing-> measuring

Correctness: Seems to be so.

Clarity: Yes

Relation to Prior Work: Please see my notes above.

Reproducibility: Yes

Additional Feedback: Post-rebuttal comments I have read the authors response and other reviews and I appreciate the effort put in by the authors in trying to address most of the concerns. As noted, the idea is interesting and well-motivated; while the experimental analysis could have been more comprehensive (e.g., my notes on the classification experiments in Section 4.3), the paper nevertheless would be a good contribution. I keep my rating unchanged.


Review 2

Summary and Contributions: Post-rebuttal update: I recommend this work put some more emphasis on the experiments in Appendix E.3: "Sensitivity to the Architecture and Regularization Strength". Some of the results could be shown in the main paper I also think the image experiment in Section 4.3 should be de-emphasized or put in the appendix. If it shall be in the main paper, clarify that the experiment is exploratory and clarify any more limitations with the application to images (in the main paper) besides the large search space I keep my rating ---------------------------------------- This paper proposes to detect interactions from neural networks via topological analysis with a method called Persistent Interaction Detection (PID). PID leverages and extends Persistent homology in topological analysis to describe the connectivity and strength of interacting features in a feed-forward neural network (FNN) on the output unit. Key to the proposed measure for interaction strength are the notions of “birth” and “death” times of connected components of interactions (interaction subgraphs of an FNN which is viewed as a graph). Here the birth and death times describe what are the start and end points of an interaction-subgraph in an FNN given decreasing lower thresholds on weight magnitudes (the lifetime of interaction subgraph is known as persistence). Interaction strength is defined as the minimal “amount of change” to eliminate an interaction subgraph, i.e. the difference between the smallest weight magnitude of an interaction subgraph and the weight that causes the “death” of the subgraph. To efficiently detect interactions, masks are used based on interaction subgraphs to focus on an interaction created at a specific neuron. Experiments are shown to outperform the state-of-the-art at 1) interaction detection where synthetic ground truth is known and 2) prediction performance via interaction encoding.

Strengths: I think this is fascinating work. Soundness: This work takes an in-depth look at how to detect feature interactions from neural networks from the perspective of topological analysis. The definition of interaction strength is principled, and the efficiency treatment is appropriate. There is also a nice discussion about the stability of PID with accompanying theory. Significance and Novelty: This work is significant and novel. Leveraging topological analysis to develop a principled approach for interaction detection is novel. Then, showing that it works compared to the state-of-the-art is a significant advancement for the interpretability of neural networks. Relevance: Interaction detection and its resulting prediction performance are becoming increasingly relevant. This method addresses this through a novel angle.

Weaknesses: I cant think of a noteworthy weakness. Good job.

Correctness: The claims and empirical methodology are correct - to my knowledge. My apologies, I didn't have enough time to carefully check the theory in the appendix.

Clarity: The paper is well written. Minor: “born” is an adjective.

Relation to Prior Work: Very clear and relevant discussions w.r.t. prior work. Good comparisons.

Reproducibility: Yes

Additional Feedback:


Review 3

Summary and Contributions: This paper proposes to detect interactions and quantify interaction strength from the topological perspective. The interaction in this paper is defined as a set of input variables that has an effect on the output of the network. Authors quantify the strength of interactions by analyzing the topological structure of the neural network. In experiments, authors conducted various experiments to show the effectiveness of the proposed method.

Strengths: It is interesting to use the algebraic topology to study the connectivity and detect the interaction in neural networks.

Weaknesses: 1. The proposed measure assumes that different dimensions of the input are well-aligned, i.e., the global interaction is independent with the specific input. However, in real datasets and applications, the mapping relationship between semantics and pixels varies among different input images. For example, give each specific input image, the pixelwise interaction depends on the image itself, instead of being pre-determined. In this case, the application of the proposed method is restricted. 2. This paper defines the connectivity of MLPs based on all paths. However, there are nonlinear operations, e.g. ReLU layers, in neural networks. These nonlinear operations will block some paths to the output, which is not considered by authors. Thus, the formulation of the proposed method seems not realistic. 3. In this paper, the interaction mainly considers edges with large weights. However, there is no experiment to show that edges with large weights can reflect all interaction effects (mot interaction effects) within the network. Authors should quantitatively prove that edges with large weights play a major role, w.r.t. effects of all other edges with not-so-large weights. In comparison, the interaction proposed by Suin Lee et al. [cite 1] can reflect all interactions modeled by the network, which is more objective than this work. 4. In Section 4.1, authors use MLPs to fit ten different functions, and detect the interaction modeled by MLPs. However, it is not always true to assume that the MLP can successfully learn the ground truth interactions encoded in the function. The MLP (if with ReLU layers) is a piece-wise linear model and cannot fit some complex functions. For example, theoretically, it is difficult for the MLP to regress y=exp(x) and y=sin(x). Thus, this experiment setting is confusing. 5. In Section 4.2, authors constructed synthetic features for each interaction, and combined these synthetic features with original inputs for training. However, at the application level, the improvement with synthetic features seems to be limited. At the interpretation level, experimental results cannot show whether the explanation is correct. The comparison is not straight forward. 6. There is no detailed introduction about the extension to the image data mentioned in section 4.3. Besides, it lacks a quantitative evaluation for the interaction detected on image data. 7. Many notations are difficult to follow. [cite 1] Lundberg, Scott, Gabriel G. Erion, and Suin Lee. "Consistent Individualized Feature Attribution for Tree Ensembles.." arXiv: 1802.03888.

Correctness: Maybe correct, but many notations are difficult to follow.

Clarity: No, many notations are difficult to follow.

Relation to Prior Work: Yes

Reproducibility: No

Additional Feedback:


Review 4

Summary and Contributions: The paper proposes a new method to identify strongly-interacting input features of a neural network. They apply filtration on the network weights to derive the persistence of subgroups of input features. The persistence of a subgroup is measured by the difference between the birth time, i.e., the weight at which all features of the subgroup is connected to the output, and the death time, i.e., the weight at which any other input feature is also connected to the subgroup. A stability theorem is provided saying that when a subgroup is detected by two networks, then the difference between its persistence is bounded by the supreme-norm difference between the two network weights. The paper also provides an algorithm to compute the strongly-interacting feature subgroups efficiently using binary matrix multiplication among the layers of the network. Experiments on synthetic data and real images are presented.

Strengths: Finding input feature interactions is an interesting and relevant problem. The algorithm and theorem are correct.

Weaknesses: The theoretical foundation of the idea is not very convincing. First of all, although the paper repeatedly stressed the connection with persistent homology, I am not convinced. Persistent homology relies on a sequence of homomorphism of homology groups through the filtration induced by inclusion. The birth and death times are well-founded as the birth and death times of topological structures (connected components, loops). However, in this paper, the structure of interest (sub-network connecting a group of input features and the output neuron) is not a well-formulated algebraic topology concept. Therefore, a lot of important theoretical property of the subject across the filtration is lost. Second, the stability theorem, although correct, is not as useful as the original stability theorem of persistent homology. In particular, the theorem only states the stability for feature sets that are detected by both networks. It is unclear whether there is high persistent feature sets in one network, but not the other. This reminds me of the fact that persistent homology’s stability is regarding critical values, but not the generators (e.g., connected components). This paper indeed focus on ``connected components’’, and thus are not well guaranteed to be stable as persistent homology. As for experiments, the synthetic experiments are interesting. But the real data experiments are not really convincing as to what the strongly-interacting input features provides that other existing interpretability approaches cannot. ** After the rebuttal period, I am willing to increase my score to 6. I trust the idea is good and useful in practice and I do believe the application of PH to neuron interaction seem to be a promising direction. However, I still have preservations on the topological analysis part. I think the method and the theorem could be improved from the following two perspectives. 1), Important details are missing as to what exact topology is captured and how is the method related to persistent homology. In the rebuttal, the authors mentioned the connection to 0D persistence through MST. But important details are still missing. The main issue to me is the constraint that the set of interactive input features need to be *connected* to the output neuron. This adds complications and it gets hard to interpret the results. For example, if you have a group of input features (a candidate set) that are connected for a long while during the filtration, but they are not connected to the final output until very late. Would you consider them persistent or not? I think both the details of connection to 0D persistence, and how to handle the challenge I just mentioned, are important and should be thoroughly explained/investigated. 2), the power of the stability results could have been improved. I suspect it is possible to create some pathological cases, in which two networks differing by a small perturbation of weights may have almost no common interaction candidate; it is possible that network 1 detects candidate set [1, 2, 3] while network 2 detects candidate set [1,2,3,4]. They are indeed very relevant but will not be considered at all in the stability theorem as it only guarantees the stability of birth/death times for a same candidate set detected by both networks. I think this is something worth diving into and better defined: for example, having a better definition of similarity between two interacting sets and have a better stability results extended accordingly.

Correctness: Yes.

Clarity: Not particularly. I think the connection with persistent homology is not really deep and convincing. To some extent, calling the underlying method a "topological analysis" is an over-statement.

Relation to Prior Work: Yes.

Reproducibility: Yes

Additional Feedback:

[Author Response · NeurIPS 2020]

We thank all four reviewers for the thoughtful comments. First, we want to point out that the novelty of the proposed **connectivity persistence** is the first metric to look at interaction strength from the topological perspective. Second, unlike existing methods, the proposed **PID** is the first interaction detection algorithm to conduct both global (4.1, 4.2) and local (4.3) interaction detection without the need to train additional interpretable model [2]. We also proved PID is insensitive to weight perturbation in models and verified its superior performance in detecting complex interactions (4.1) and in engineering features which greatly boosted the performance of models in real-world tasks (4.2). We believe this is an important contribution for bringing topological properties and interpretability to interaction detection. Below, we address the reviewers' comments individually:

**Discussion about NID and PID (R1: Q1.1).** To extract interactions, PID considers connectivity of the entire NN. In contrast, NID leverages weights beyond the first hidden layer to obtain the maximum gradient magnitude of the hidden units in the first hidden layer, loosing some information encoded in latter layers in the process. Hence, the similar results of NID and PID are likely because the latter layers played lesser roles in this specific setting. However, we remark PID constantly outperformed NID with various settings, as shown in Appendix E.3, Figure 8, 9, and 10.

**Discussion about AG (R1: Q1.2).** We remark that the results of AG is adapted from NID [7], which attributes AG's performance difference in $F_5, F_6, F_8$ in "limitations on the model capacity of AG, which is tree-based".

**Image Experiment (R1: Q2).** CNNs are indeed trained to classify images. As it is standard to build saliency maps to evaluate how CNNs make decisions, we aggregate interaction strengths of interacting pixels detected by PID to get the importance of each pixel on the image (line 321). We remark that a key difference between interaction detection and explainable CV (e.g., GradCAM) is that the latter does not consider interactions between pixels because it does not have access to Hessian matrix. In contrast to PID, explainable CV cannot give strength between any group of pixels. For ImageNet, our PID has a $2^{224\times224} \approx 10^{1021609}$ search space (the search space for MNIST is $10^{236}$), which is intractable. To illustrate the search space's magnitude, the search space of AlphaGO is $10^{360}$ [6].

**Tasks other than classifications (R1: Q3)** We will include the discussion in the revised manuscript.

**Inadequate broader impact (R2).** The main application of global interaction detection is knowledge discovery. Therefore, PID can help us discover the combined effects of drugs on human body. For example, by utilizing PID on patients' records, we might find using Phenelzine togther with Fluoxetine has a strong interaction effect towards serotonin syndrome. Thus, PID has great potential in helping the development of new therapies for saving lives.

**PID assumes well-aligned data & nonlinear operations block paths (R3: Q1, Q2).** We remark PID is agnostic to input alignment. See 4.1 and 4.2 for global interaction detection on tabular (well-aligned) data and 4.3 for local interaction detection on image (not well-aligned) data. Appendix D addresses how to adapt PID for local interaction detection by incorporating nonlinear operation (ReLU).

**Mainly considers edges with large weights (R3: Q3).** We remark that PID extracts interaction based on persistence, not large weight (Section 3). In addition, [1] can only capture "pairwise interaction effects", not all interactions.

**MLP cannot fit some complex function (R3: Q4).** According to the Universal Approximation Theorem, MLP (with ReLU) can fit any continuous function. Appendix E.1 shows $\exp(\cdot)$ is considered in $F_3, F_4, F_5, F_6$, and trigonometric functions are considered in $F_6, F_8, F_{10}$. Appendix E.3 shows the test error of trained MLPs is very low.

**Limited improvement with real world data (R3: Q5).** For tabular data, we remark an improvement around 0.001 in AUC on these datasets is considered SOTA [3]. In addition, 4.2 shows interactions detected by PID are useful to real-world tasks. Since human-found interactions have been setting the standard in the industry, showing PID finds interactions matching those found by humans is meaningful.

**Introduction for image task (R3: Q6).** For introduction to the image task, please see R1: Q2. We also remark it is conventional to evaluate interaction detection task on image data qualitatively [2].

**Difficult notations (R3: Q7)** We apologize and will modify our mathematical notations in the revised manuscript.

**Persistent homology (R4: Q1).** We are aware that some nice properties have been lost, such as Excision Theorem does not hold. However, as NNs contain only 1-simplex, many of these properties degenerate to the field of graph theory and become easier to evaluate whether they are useful for interaction detection. Due to the page limit, here we only list our high-level idea. From the graph theory perspective, the proposed filtration process is equivalent to building maximum spanning trees (MSTs) of NNs using Kruskal algorithm. The proposed persistence of feature groups is the gap length between MSTs of two sub-networks. There are many papers discussing about the relationship between MSTs and persistent homology, and we could easily extend their results [4, 5]. We will add it in the revised manuscript and discuss about the limitations accordingly. By extending the Barcode from persistent homology and $\langle \phi = \lambda \rangle$-connection from size theory, we derived a topology-motivated algorithm to efficiently detect interaction (Lemma 1) with stability guarantee (Theorem 1).

**Unclear whether there is high persistent feature sets in one network, but not the other (R4: Q2).** The proof that this situation only happens if the perturbation magnitude $\delta$ is greater than a threshold relating to persistence will be added to the revised manuscript. We remark that the stability theorem in the paper is customized for Algorithm 1, which is not comparable to the stability theorem of topological features in persistent homology. Also, we clarify that the results in Appendix C, Table 3 actually took this situation into account. Namely, if an interaction $\mathcal{I}$ is only detected in $f$ but not in $g$, we treat $\rho_g(\mathcal{I}) = 0$ to get perturbation results.

**Unique contribution of interactions detected by PID (R4: Q3).** We will add comparisons in the revised manuscript.

[1] Lundberg, S. M., Erion, G. G., & Lee, S. I. (2018). Consistent individualized feature attribution for tree ensembles. In *arXiv preprint arXiv:1802.03888*.
[2] Tsang, M., Cheng, D., Liu, H., ... & Liu, Y. (2020). Feature Interaction Interpretability: A Case for Explaining Ad-Recommendation Systems via Neural Interaction Detection. In *ICLR* 2020
[3] Khurana, U., Samulowitz, H., & Turaga, D. (2017). Feature engineering for predictive modeling using reinforcement learning. In *arXiv preprint arXiv:1709.07150*.
[4] Steele, J. M. (1988). Growth rates of Euclidean minimal spanning trees with power weighted edges. In *The Annals of Probability, 1767-1787*.
[5] Robins, V., ...& Bradley, E. (2000). Computational topology at multiple resolutions: foundations and applications to fractals and dynamics (*Doctoral dissertation, University of Colorado*).
[6] Silver, D., Huang, A., Maddison, C. J., Guez, ...& Dieleman, S. (2016). Mastering the game of Go with deep neural networks and tree search. In *nature*, 529(7587), 484-489.).
[7] Tsang, M., Cheng, D., & Liu, Y. (2017). Detecting statistical interactions from neural network weights. *arXiv preprint arXiv:1705.04977*.


[Meta-Review · NeurIPS 2020]

All knowledgeable referees have confirmed the novelty parts and contributions of this work. I recommend acceptance of this paper and suggest the authors refine the paper before publication. Particularly, the concerns and suggestions raised by R#4 & R#3 should be addressed. AC and SAC discussed this paper on the issues raised by R3 and converged to accept. The authors are encouraged to discuss whether CNN belongs to the models that can be explained by the method proposed.